 TRANSPARENT PROCESS  OPEN ACCESS

# Quantitative analysis of protein interaction network dynamics in yeast

Albi Celaj[1,2,3,†] , Ulrich Schlecht[4,5,†], Justin D Smith[4,6] , Weihong Xu[4] , Sundari Suresh[4,5], Molly Miranda[4,5], Ana Maria Aparicio[4,5], Michael Proctor[4,5], Ronald W Davis[4,5,6], Frederick P Roth[1,2,3,7,8,*]  & Robert P St.Onge[4,5,**] 

## Abstract

Many cellular functions are mediated by protein–protein interaction networks, which are environment dependent. However, systematic measurement of interactions in diverse environments is required to better understand the relative importance of different mechanisms underlying network dynamics. To investigate environment-dependent protein complex dynamics, we used a DNA-barcode-based multiplexed protein interaction assay in *Saccharomyces cerevisiae* to measure *in vivo* abundance of 1,379 binary protein complexes under 14 environments. Many binary complexes (55%) were environment dependent, especially those involving transmembrane transporters. We observed many concerted changes around highly connected proteins, and overall network dynamics suggested that "concerted" protein-centered changes are prevalent. Under a diauxic shift in carbon source from glucose to ethanol, a mass-action-based model using relative mRNA levels explained an estimated 47% of the observed variance in binary complex abundance and predicted the direction of concerted binary complex changes with 88% accuracy. Thus, we provide a resource of yeast protein interaction measurements across diverse environments and illustrate the value of this resource in revealing mechanisms of network dynamics.

**Keywords** environmental response; mRNA expression; network dynamics; protein complementation assay; protein–protein interactions

**Subject Categories** Genome-Scale & Integrative Biology; Methods & Resources; Network Biology

**Mol Syst Biol.** (2017) 13: 934

## Introduction

The molecular function and cellular role of a protein often cannot be understood without knowledge of its interactions with other proteins. For this reason, multiple high-throughput methods have been developed to identify direct protein–protein interactions (PPIs) on a genomewide scale, for example, using yeast 2-hybrid (Y2H) or protein-fragment complementation assay (PCA) methods. These and complementary techniques for detecting co-complexation, for example, affinity purification coupled with tandem mass spectrometry (AP-MS), have yielded a wealth of PPI data in model organisms and humans (Uetz *et al*, 2000; Ito *et al*, 2001; Butland *et al*, 2005; Stelzl *et al*, 2005; Gavin *et al*, 2006; Krogan *et al*, 2006; Tarassov *et al*, 2008; Yu *et al*, 2008). A positive Y2H assay suggests that two expressed proteins are capable of a direct biophysical PPI in the context of the *Saccharomyces cerevisiae* nucleus, but whether, where, and when this interaction is physiologically relevant is left undetermined. In contrast, various PCA approaches have been developed which test for an interaction in the native cellular environment at native expression levels (Tarassov *et al*, 2008; Schlecht *et al*, 2012). These approaches, reportedly able to detect as few as 25 complexes per cell (Remy & Michnick, 1999), can directly measure the dynamic dependence of a PPI on growth environment.

While the results of many high-throughput PPI assays are interpreted as static maps of physical connections, it is known that the variability of protein complexes and the coordinated dynamics of physically linked gene products underlie fundamental aspects of cellular function. In an attempt to capture this, static protein–protein interactomes have been used as a "scaffold" on which to overlay and interpret other genome-scale data such as gene expression and metabolic fluxes (Ideker *et al*, 2002; Luscombe *et al*, 2004; Sauer, 2004; de Lichtenberg *et al*, 2005). These approaches have shown the value of PPIs in contextually understanding gene function, but they cannot straightforwardly identify

1 Departments of Molecular Genetics and Computer Science, University of Toronto, Toronto, ON, Canada
2 Donnelly Centre, University of Toronto, Toronto, ON, Canada
3 Lunenfeld-Tanenbaum Research Institute, Sinai Health System, Toronto, ON, Canada
4 Stanford Genome Technology Center, Stanford University, Palo Alto, CA, USA
5 Department of Biochemistry, Stanford University School of Medicine, Stanford, CA, USA
6 Department of Genetics, Stanford University School of Medicine, Stanford, CA, USA
7 Canadian Institute for Advanced Research, Toronto, ON, Canada
8 Center for Cancer Systems Biology, Dana-Farber Cancer Institute, Boston, MA, USA
*Corresponding author. Tel: +1 416 946 5130; E-mail: fritz.roth@utoronto.ca
**Corresponding author. Tel: +1 650 721 2976; E-mail: bstonge@stanford.edu
†These authors contributed equally to this work

quantitative changes in PPI complex levels, nor directly determine protein complex levels in an environment-dependent cellular state.

A simplified view of PPI dynamics as "binary switches" in which interactions are either present or absent is common and can offer biological insights (de Lichtenberg et al, 2005; Greene et al, 2015) but ignores potentially important quantitative changes in protein complex abundance. The proliferating cell nuclear antigen (PCNA) complex serves to illustrate the idea of quantitative environmentally responsive protein interactions. PCNA, a major factor in DNA replication and repair, forms a chromatin-bound complex with other proteins at sites of DNA damage in response to gamma irradiation in a dose-dependent manner (Balajee & Geard, 2001; Mailand et al, 2013). In another example, salt stress leads to in vivo activation of the HOG (high-osmolarity glycerol) signal transduction pathway, which is quantitatively dependent on the interaction between Sho1 and Pbs2 (Marles et al, 2004).

In order to address the limitations of static interactome maps, multiple studies have begun to identify condition-specific PPI changes directly. We have previously implemented highly multiplexed murine dihydrofolate reductase (mDHFR)-based PCA, which can detect changes in the abundance of hundreds of binary protein complexes in parallel. In this approach, the mDHFR fragments are fused to genes at their genomic locus under control of the endogenous promoter, thus allowing an in vivo study of binary protein complex level changes. This approach was used to examine the effects of 80 small molecules on 238 yeast binary protein complexes, uncovering multiple positive and negative chemical modulators (Schlecht et al, 2012). A related study by Rochette et al (2014) used a plate-based mDHFR PCA to test the response of 1,338 yeast protein interactions to methyl methanesulfonate, an alkylating agent that induces DNA damage. This study identified PPI changes in diverse cellular processes and found that, in the DNA damage response, protein relocalization is a major driver of PPI changes. A strength of the mDHFR PCA in measuring dynamic interactions is that quantitative changes in relative strain abundance reflect quantitative changes in binary protein complex abundance (Schlecht et al, 2012; Freschi et al, 2013), allowing for a detailed view of PPI remodeling by simply measuring cellular growth rates.

Here, we extend our previously developed multiplex barcoded mDHFR PCA (BC-PCA) to investigate the effects of 14 chemical and environmental perturbations on 1,379 binary protein complexes. We observed widespread PPI changes in these conditions, many of which were informative for the specific stimulus applied. Altered PPIs tended to concentrate in large subnetworks and were often centered around highly connected hub proteins. More closely examining the shift from fermentative to respiratory growth, we found a highly significant relationship between binary protein complex abundance and mRNA levels that explains a large portion of the observed network dynamics. This correlation was predictive, suggesting that reasonably accurate first-order estimates of PPI network dynamics can be made using only mRNA data.

## Results

### Construction of a genome-scale barcoded protein-fragment complementation assay

In the murine dihydrofolate reductase protein complementation assay (mDHFR PCA), two proteins of interest are fused to two respective fragments of mDHFR. Upon successful physical interaction of the two target proteins, the mDHFR fragments fold together into the native conformation and give rise to a functional enzyme that is resistant to methotrexate (MTX). Interaction-dependent reconstitution of MTX-resistant mDHFR allows for growth-based selection, where the extent of MTX resistance is dependent on the intracellular concentration of the PPI complex. The majority of all possible binary PPI combinations in the yeast genome have been previously subjected to mDHFR testing, leading to the identification of 2,770 PPIs under a set of standard laboratory conditions (i.e., growth at 30°C on solid media containing glucose, a rich nitrogen source, and all essential supplements; Tarassov et al, 2008).

Having established the multiplex BC-PCA assay at a smaller scale (Schlecht et al, 2012), here we attempted to scale it to as many known PCA interactions as possible in a pooled and barcoded format. We first reconstructed 2,394 (see Dataset EV1) of the 2,770 Tarassov et al (2008) strains, 1,701 of which were verified to grow in liquid minimal media containing MTX (Fig EV1A and B). We observed a broad range of growth rates among these strains. This likely reflects strain-specific differences in the abundance of reconstituted mDHFR complexes that arise from differences in abundance of the binary protein complex of interest, but could also reflect differences in DHFR reconstitution efficiency arising from steric effects of different proteins fused to mDHFR fragments. Consistent with a previous study (Freschi et al, 2013), we found growth rate to be significantly correlated with the protein expression level of the least abundant protein (using abundance data from Wang et al, 2012) in the interacting pair (Pearson's $r = 0.31$, $P < 2.2e-16$, Fig EV1C). Indeed, this correspondence is theoretically expected in cases where PPI affinity is high and each pair of interacting proteins is independent of others (see Materials and Methods). Also consistent with this idea, we found that the subset of confirmed PPI pairs from Tarassov et al (2008) tended to have least abundant proteins that were more abundant than the least abundant proteins of unconfirmed PPI pairs (Fig EV1D). Taken together, these results support the idea that quantitatively measuring MTX resistance of these PCA strains can capture protein complex abundance information.

To adapt the successfully recreated PCA strains to the BC-PCA assay, barcode cassettes were transformed into corresponding F [1,2]-containing haploid strain for each pair, which was then mated with the corresponding F[3]-containing partner (Materials and Methods, Fig EV1A) to create a barcoded diploid strain. We successfully incorporated 1,432 barcodes, representing 1,428 of 1,701 unique interactions (Dataset EV1). These strains were then pooled and competitively grown in selective media. Genomic DNA was isolated from an aliquot, and barcodes were PCR-amplified and hybridized to a high-density oligonucleotide array in six replicates. Most barcodes (1,383 of 1,432; representing 1,379 unique interactions) were detected across all six replicates, and the rest were excluded from further analysis. As was previously found, microarray signal intensity values showed very high correlation among six replicates ($r \geq 0.97$, Fig EV1E), and correspondence to noncompetitive growth rates measured in isogenic culture ($r = 0.69$, Fig EV1F). This confirms that our pool-based assay provides a reproducible quantitative growth measure for each individual strain.

## Identifying protein complex dynamics under diverse conditions

Many previous studies have mapped and analyzed "static" PPI networks, in which the interaction assay reveals protein pairs that are capable of interacting if expressed at sufficient concentrations at the same time and place. However, the question of how and to what extent PPI complex levels vary across different conditions has not been studied at a large scale. Given that the BC-PCA assay yields quantitative strain abundance measurements (across a large dynamic range) which correspond to growth, and because growth directly relates to reconstituted mDHFR abundance (Remy & Michnick, 1999), BC-PCA represents a means of addressing this question. Using BC-PCA, we can detect a condition-specific deviation of growth rate in response to a perturbation (compared to a "reference" state) and infer a change in binary protein complex levels.

To represent broad classes of environmental change, we chose a set of 14 different perturbations, including: addition of small molecules (e.g., FK506, atorvastatin, doxorubicin); altered nutrient composition of the growth medium (e.g., ethanol instead of dextrose as the sole carbon source, nitrogen starvation, addition of specific amino acids); and abiotic stress conditions (e.g., growth at high temperature, in a high salt concentration, or under oxidative stress). Dataset EV1 lists details of all growth environments. All experiments were performed in both *selective* and *non-selective* (i.e., without MTX) media, with the latter acting as a control to identify and exclude cases where the incorporation of the mDHFR tag had an impact on growth in the condition tested (Rochette *et al*, 2014; Dataset EV2). In most conditions, significant growth changes in *non-selective* media were observed in < 2% of all strains in the pool (see Materials and Methods). Such strains were specifically excluded from the conditions in which they were identified, as were additional strains containing PCA fragments appearing multiple times in excluded strains (Dataset EV2).

For each PPI in each condition tested, we calculated $R$, the ratio of barcode abundance in that condition (as measured by microarray signal intensity) to barcode abundance in the reference condition (selective dextrose-containing media plus 1% DMSO). Requiring both the UP- and DOWN-tag of each strain to meet our significance and effect size thresholds ($q$-value < 0.05 and $\log_2(R) > 0.25$ for accumulated, and $q$-value < 0.05 and $\log_2(R) < -0.25$ for depleted interactions), we found 757 binary complexes that varied in at least one condition (Dataset EV2). Hierarchical clustering of these data showed that all experiment replicates were grouped as closest neighbors (Fig 1A), indicating a reproducible assay. We examined the reproducibility of a subset of candidate dynamic binary complexes using isogenic growth experiments and found a strong correlation between growth assays and pooled barcode fluorescence intensity values in most cases ($r > 0.6$, Fig EV2A and Dataset EV1). We also observed binary complexes where there was a clear dose-dependent relationship between the addition of a small molecule and the relative growth rate under MTX selection (Fig EV2B). Together, these results suggest that the BC-PCA assay can measure quantitative changes in abundance of many binary complexes, supporting its utility for studying condition-dependent global PPI remodeling.

We clustered dynamic interactions based on their pattern of change across environments. We observed several subnetworks that shared a common protein (Fig 1B), suggesting that control of the common protein may have led to concerted changes in binary complexes containing that protein. For example, Fmp45 was a member of several binary complexes that accumulated during respiratory growth (using ethanol as a carbon source) and under heat and high-salt stress (Fig 1B). Similarly, the addition of hydrogen peroxide led to the accumulation of binary complexes containing the Ftr1 protein, a high-affinity iron permease (Stearman *et al*, 1996). Finally, growth in methionine-supplemented media led to the depletion of binary complexes containing the methionine permease Mup1. This cluster also contained other proteins important for methionine metabolism, underscoring that the function of responsive complex components was logically connected to environmental change.

We observed many binary complexes that changed between conditions, but interestingly, within each condition there was a comparable number of accumulated and depleted binary complexes (Fig 1C and Dataset EV2). Conditions that induced the most widespread changes were the shift from fermentation to respiration (i.e., dextrose to ethanol), the addition of amino acids to the media, and growth at high temperature, while the addition of bioactive small molecules such as FK506 and atorvastatin resulted in a much smaller number of altered binary complexes (Fig 1C). These results may be explained by the fact that pharmaceuticals are often selected for specificity (i.e., to minimize off-target and side effects). In contrast, metabolic shifts or exposures to abiotic stress factors can induce widespread cell-physiological effects (Gasch *et al*, 2000) and regulatory responses that may have evolved given frequent exposure to similar environments in the evolutionary history of yeast (Gasch & Werner-Washburne, 2002; Gasch, 2007). Contrary to this pattern, the pharmaceutical agent doxorubicin resulted in many protein interaction changes (Fig 1C), possibly due to widespread non-specific effects of DNA damage and corresponding induction of a DNA damage response (Westmoreland *et al*, 2009).

## Dynamic binary complexes exhibit condition-dependent functional trends

About half (757) of the binary complexes tested were dynamic in at least one condition. Most (672) of these dynamic binary complexes were found to be specific to a few (from one to three) conditions, while only 86 binary complexes were frequently dynamic (exhibiting change in four or more conditions; Fig 2A). To assess functional trends among gene products participating in frequently dynamic binary complexes, we used the FuncAssociate web server (Berriz *et al*, 2003; Berriz *et al*, 2009, see Materials and Methods) to determine over-represented Gene Ontology (GO) terms. This revealed that gene products involved in frequently dynamic binary complexes were enriched for *plasma membrane* localization ($q <$ 1e-04), and for *active transmembrane transporter activity* ($q =$ 0.012; Dataset EV3). Among the proteins involved in frequently dynamic binary complexes were members of the 12-span drug:H(+) antiporters (Tpo1, Tpo2, Tpo3), multidrug ABC transporters (Pdr5, Pdr12, Snq2, Yor1), and transporters for glucose, iron and methionine (Hxt1, Hxt2, Hxt5, Ftr1, Mup1). Two proteins of unknown function (Ybl029c-A and Ina1/Ylr413w) were also found to take part in frequently dynamic interactions, suggesting a potential role in responding to environmental stress (Fig 2B).

To further assess functional trends among dynamic binary complexes, we again used GO enrichment analysis to determine

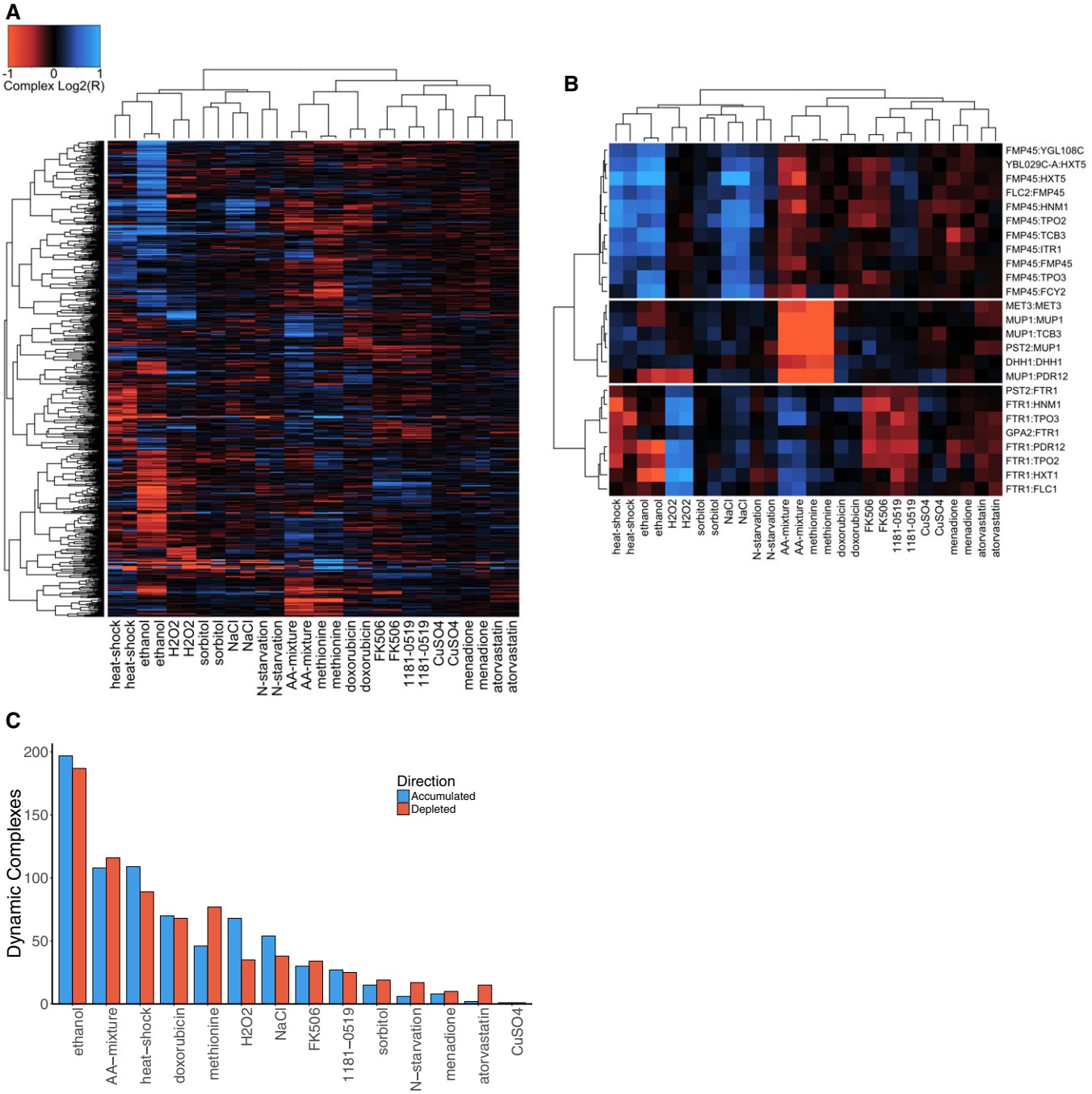

**Figure 1.  Environmental cues elicit changes in the yeast protein interaction network.**

A   Heatmap depicting $\log_2$-ratios of 757 binary protein complexes that displayed a significant change in at least one of the conditions tested here. Complexes are arranged on the $y$-axis, and conditions are arranged on the $x$-axis. Accumulated and depleted signals are colored in blue and red, respectively. Dendrograms on the left and on top show clustering of complexes and samples, respectively.

B   Three clusters of binary protein complexes. Complexes are labeled on the right.

C   Barplot depicting the number of protein complexes whose abundance significantly changed ($\log_2(R) > 0.25$ or $\log_2(R) < -0.25$ for both UP- and DOWN-tags in the same direction, $q < 0.05$ for both UP- and DOWN-tags) in response to 14 perturbations (indicated on the $x$-axis). Number of complexes that were accumulated or depleted in response to each condition are shown in blue and orange bars, respectively.

which functions were enriched among gene products participating in either accumulated or depleted binary complexes, examining each condition separately (Figs 2C and EV3). Under many

conditions, transmembrane transporters and plasma membrane proteins were over-represented among gene products participating in dynamic binary complexes. In some cases, binary complexes

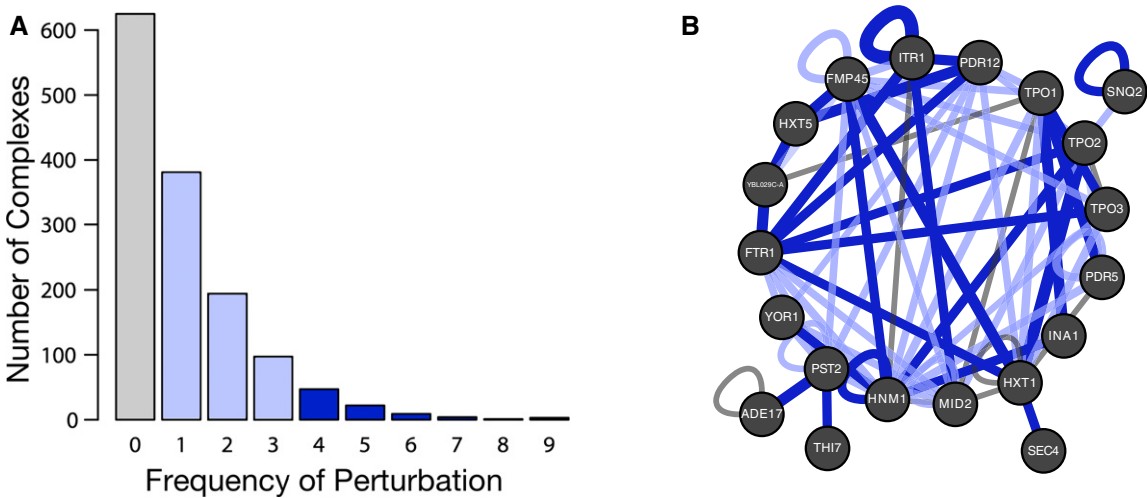

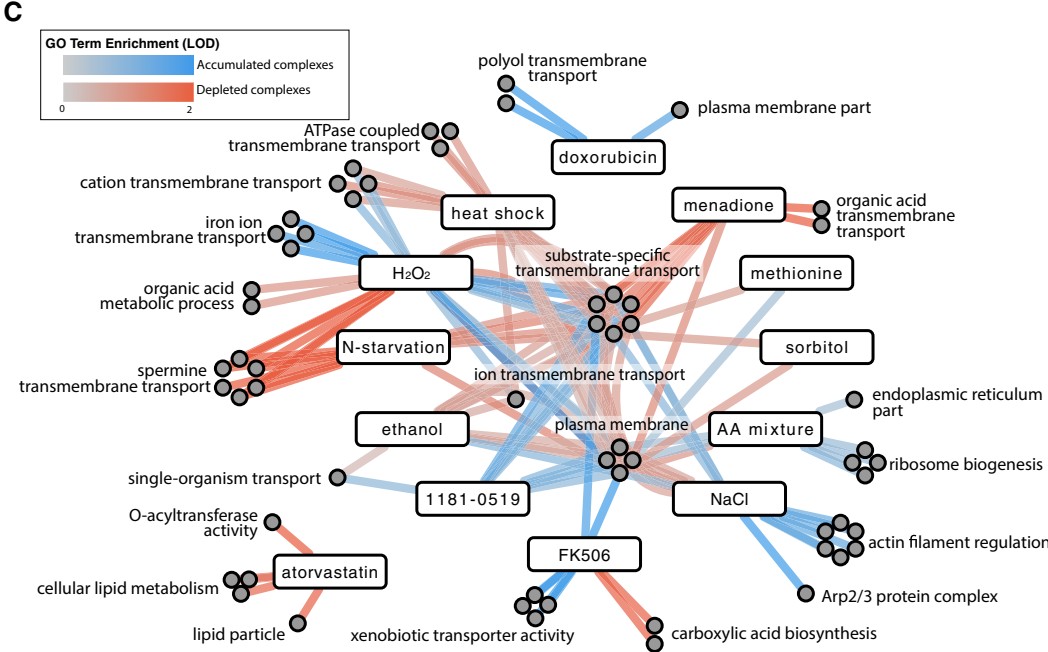

**Figure 2.  Functional trends of dynamic complexes.**

A    Barplot showing the frequency at which each binary protein complex was found to be dynamic (i.e., accumulated or depleted). The number of complexes ($y$-axis) is plotted against the number of conditions they responded to ($x$-axis). Colors indicate complexes that were dynamic in 0 (gray), 1–3 (pale blue), and 4 or more (dark blue) conditions.

B    Network illustrating plasma membrane complexes that were frequently modulated. Proteins are indicated as nodes. Interactions that are frequently perturbed (i.e., four or more times) are shown with dark blue edges. Other interactions are indicated by semi-transparent edges—interactions perturbed 1–3 times are in pale blue and static interactions are in gray. Edge width is proportional to perturbation frequency.

C    Network illustrating functional trends of genes participating in dynamic complexes. For each condition, we assessed functional enrichment among gene products participating in either accumulated complexes (blue) or depleted complexes (red). Lines connect terms to their respective condition(s) and are shaded by degree of GO term enrichment measured using the $\log_{10}$-odds score (LOD). Terms here have been manually grouped and summarized. All original terms are available in Fig EV3 and Dataset EV3.

changing in a condition were clearly related to the environment. For example, there was an enrichment of "xenobiotic transporter activity" for proteins involved in accumulated complexes under the small-molecule drug FK506 ($q = 0.0023$). Here, the enrichment phenomenon was driven by previously observed binary complex

changes involving the Snq2, Pdr5, and Yor1 efflux pumps (Schlecht *et al*, 2012).

In other cases, the changes were less intuitive but consistent with previous findings. For example, binary complexes that accumulated in the NaCl condition were enriched for proteins in the Arp2/3 actin

polymerization complex. This is consistent with previous studies finding that overall actin organization is responsive to osmotic stress (Chowdhury *et al*, 1992), and the genes driving this GO term enrichment (ARC40, ARP2, ARC15, ARC19, ARC18, and ARC35) have all been previously shown to have elevated transcript levels under hypersaline conditions (Berry & Gasch, 2008). In another example, binary complexes depleted under atorvastatin were enriched for proteins involved in the lipid biosynthetic process. This is consistent with atorvastatin's role as an inhibitor of HMG-CoA reductase (Endo, 1992; Leszczynska *et al*, 2009).

### Interaction network dynamics are largely driven by "protein-centric" changes

In every condition tested, we sought to identify patterns of variability in the protein network. Observing that some clusters of binary protein complex abundance changes involve a common protein (Fig 1B), we further explored the topology of binary protein complex dynamics. In some cases, large subnetworks which are fully connected by dynamic interactions in the same direction were formed (Fig 3A shows an example from the doxorubicin condition). Frequently, these subnetworks ("dynamic components") were centered on highly connected "hub" proteins (defined here as proteins with 10 or more interactions; see Dataset EV4 for a list of the 74 hubs identified).

Sets of interactions around a given hub protein were often "concerted" in that they changed in the same direction. For example, the hub proteins Ade17, Hxt1, Ftr1, and Ppz1 all exhibit apparently concerted binary complex level changes in the doxorubicin environment (Fig 3A). We sought to systematically identify hubs exhibiting concerted changes (Fig 3B; Materials and Methods). This analysis revealed that 50 hubs were concerted in at least one condition, representing 68% of all hubs (Dataset EV4). This included the examples shown in Fig 1B, wherein the sets of binary complexes centered on Fmp45, Mup1, and Ftr1 each exhibited concerted change under multiple perturbations.

Concerted binary complex changes around a protein are most simply explained by a change in the general availability of that protein for complex formation. For example, changes in the abundance of a given protein should, by mass action, tend to alter the abundance of all binary complexes in which it participates and do so in the same direction in each case. We therefore refer to patterns of concerted binary complex changes as "protein-centric" effects. Conversely, if changes around a protein are not concerted, this may be explained by mechanisms which modulate interactions independently, for example, through a post-translational modification at an interaction interface, or through direct interference or facilitation of an interaction by a small molecule. We refer to patterns of independently modulated interaction changes as "interaction specific".

Given that we observed many concerted binary complex changes (Fig 3B), we wished to explore the extent to which the global patterns of binary complex changes could be explained by protein-centric as opposed to interaction-specific dynamics. Depending on the relative contribution of these two mechanisms, dynamic subnetworks should exhibit a different overall topology. Specifically, we expect dynamic subnetworks resulting from purely protein-centric changes to be grouped into distinct network areas, yielding larger mutually connected subgraphs ("component sizes"), as well as a higher density of interactions within dynamic subnetworks ("subgraph density") than we would expect to observe if dynamics were driven by many independent interaction-specific changes. Dynamic subnetworks affected by a combination of protein-centric and interaction-specific changes are expected to exhibit intermediate component sizes and subgraph densities.

In order to explore what proportion of protein-centric dynamics are most consistent with the patterns of observed binary complex changes, we generated dynamic networks *in silico* by randomly selecting interactions from our entire BC-PCA network (see Materials and Methods). For each given growth condition, we simulated networks having the same number of enriched, depleted, and unchanged interactions that were observed in the BC-PCA assay. The random selection procedure was varied to yield different proportions of protein-centric and interaction-specific effects. For each selection procedure, we then empirically estimated how often, among 1,000 random networks, simulations yielded a dynamic subnetwork with a topology similar to that of the observed dynamic subnetwork. As measures of dynamic network topology, we used both the size of the largest component and the interaction density.

We first simulated dynamic networks with purely interaction-specific changes for all nine conditions that produced a large number of changed binary complexes (set to ≥ 50), and found that none of the largest connected component sizes were consistent with a purely interaction-specific scenario (Fig 3C, bottom row). We next simulated purely protein-centric binary complex changes for all conditions (Fig 3C, top row) and also simulated networks with different mixtures of interaction-specific and protein-centered dynamics. The largest component sizes of all nine (100%) of the conditions were consistent with a purely protein-centered model. The proportions of protein-centered changes most consistent with the observed largest component size ranged between 40 and 100% across conditions (Fig 3C). Evaluating dynamic subgraph density

**Figure 3. Environmental perturbations elicit a mixture of protein-centric and interaction-specific changes.**

A    Network illustrating the largest connected component of complexes, which were accumulated (bottom) or depleted (top) in the presence of doxorubicin. Incidentally, TPO2 is part of both subnetworks. Proteins are indicated as nodes; interactions are depicted as edges (accumulated in blue, depleted in red).

B    Heatmap illustrating whether complex abundance changes associated with a given hub protein (*y*-axis) had a bias (*q* < 0.05) toward depletion (red) or accumulation (blue) in each condition (*x*-axis).

C    Comparison of the largest components found in simulated networks to those observed in the BC-PCA data. For the indicated conditions (*x*-axis), the observed number of accumulated, depleted, and unchanged interactions was used to generate random networks based on a "protein-centric" model, an "interaction-specific" model, and combinations thereof (*y*-axis; see Materials and Methods). Each square summarizes the results from 1,000 simulations (see legend). The insets expand and illustrate two simulation scenarios under the "AA-mixture" condition; histograms show the simulation results for accumulated (left) and depleted (right) interaction subnetworks, under purely "protein-centric" (top) and purely "interaction-specific" (bottom) models. The observed BC-PCA result is represented by a red line.

D    As in (C), comparing the subgraph density of simulated networks to those observed in the BC-PCA data.

yielded similar conclusions about the importance of protein-centric changes (Fig 3D)—again, none of the nine conditions were consistent with a purely interaction-centric model. However, four of the nine conditions required at least some interaction-specific changes

(10%) to yield a subgraph density consistent with the BC-PCA results. Thus, for each of the measures we used, protein-centered dynamics were important for explaining the observed dynamic subnetwork topology.

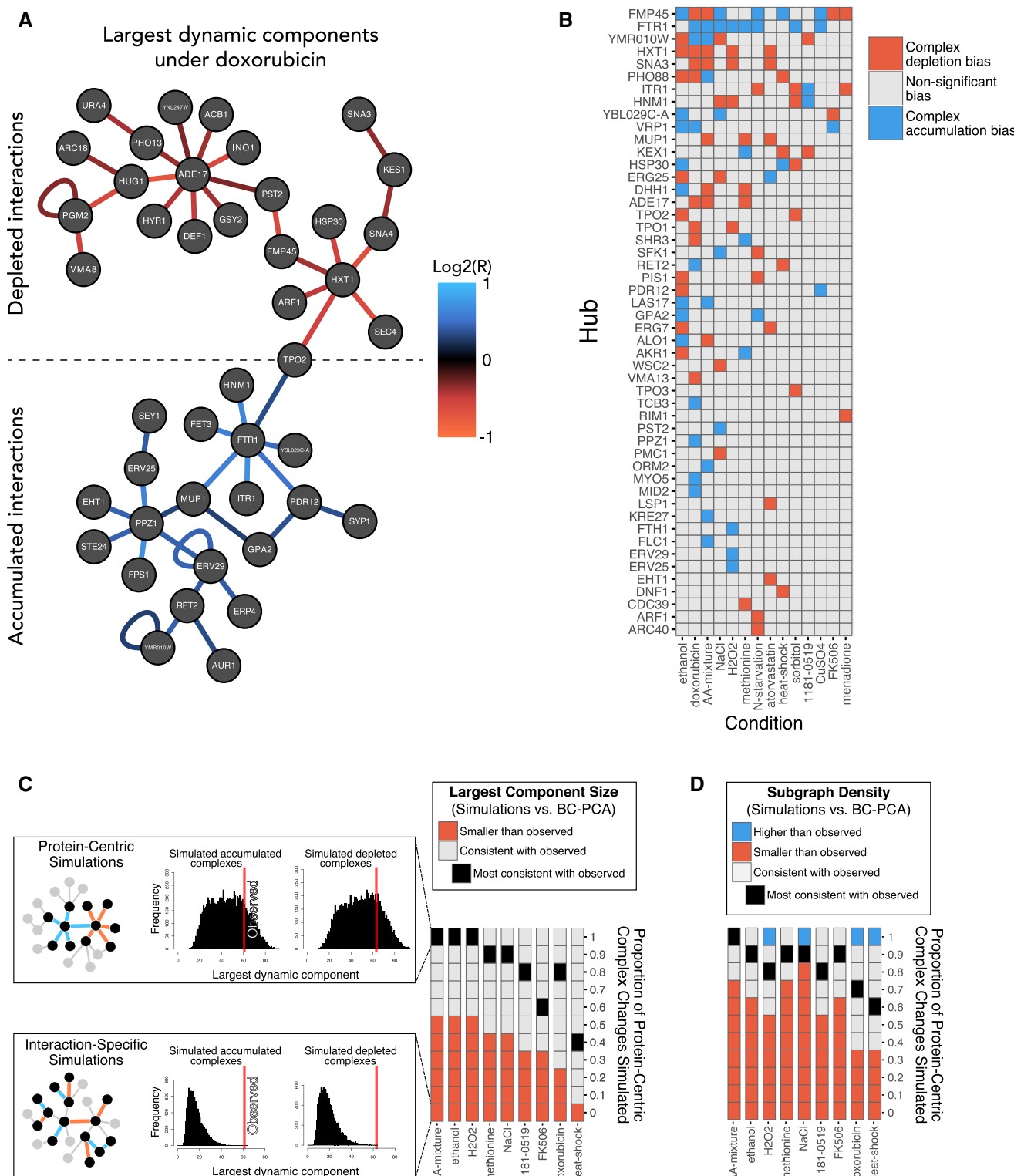

**Figure 3.**

These results do not identify the mechanism of interaction regulation (e.g., modulation of transcription, localization, direct small-molecule interference, or post-translational modification), nor do they specifically determine which dynamic complexes are the result of interaction-specific compared to protein-centric effects. Moreover, different environmental stresses may yield effects different from those we observed. Nevertheless, the frequent observation of dynamic interactions centered on hubs suggests the prevalence of protein-centric interaction dynamics, in which interaction network dynamics are substantially driven by regulation of a few proteins.

## Transcriptome changes predict binary complex abundance changes

We next investigated the extent to which concerted protein-centric interaction dynamics were the consequence of transcriptome changes. The potential impact of transcriptional change on specific interactions is well known. For example, we have previously shown that transcriptional up-regulation of the *PDR5* and *SNQ2* genes accompanies the accumulation of several drug-efflux pump homo-dimers (such as Pdr5:Pdr5 and Snq2:Snq2) in response to FK506 (Schlecht *et al*, 2012). Indeed, mRNA expression data have previously been combined with "static" protein–protein interaction networks to predict which interactions are present in a given condition (de Lichtenberg *et al*, 2005; Greene *et al*, 2015). However, these previous studies made no attempt to quantitatively predict and experimentally validate the changes in protein complex levels that result from transcriptional change. We interrogated the relationship between transcriptional changes and binary complex dynamics *in vivo*.

A relationship between expression levels and PPI complex abundance has been previously modeled based on mass action (Maslov & Ispolatov, 2007); however, the validity of this model has not been experimentally confirmed. To investigate this relationship further, we more closely examined the shift from fermentation to respiration (i.e., growth in glucose medium versus growth in ethanol medium), as the respiratory growth condition yielded the greatest number of binary protein complex abundance changes (nearly 400, Fig 1C). Yeast cells are known to undergo widespread transcriptional changes during a diauxic shift (DeRisi *et al*, 1997; Gasch *et al*, 2000). To obtain relative mRNA expression data, cells were grown in selective media supplemented with dextrose as a pre-culture and then shifted to selective media containing ethanol as the sole carbon

source, with samples taken at 0, 0.5, 1, 4, and 12 h after the transfer (see Dataset EV5).

We sought to use relative mRNA levels in an approximate ("first-order") model that could predict relative binary protein complex levels, using principles from the law of mass action. We made several simplifying modeling decisions, including: (i) modeling relative protein levels as moving in concert with relative mRNA levels; (ii) modeling binary protein complex dynamics in isolation, for example, such that involvement of a protein in one complex did not deplete its availability for another; (iii) treating each interaction as occurring at relatively high affinity; (iv) using an aggregate of measurements in previous studies to estimate the baseline concentration of each protein (see Materials and Methods). Although each of these modeling decisions should be revisited in more advanced models, we found that predictions of relative binary complex abundance using this "first-order" model yielded striking correspondence with the observed growth changes observed in BC-PCA. The highest correlation was observed at 4 h after the shift to ethanol ($r = 0.46$, Figs 4A and EV4), and these correlations were not evident in non-selective media ($r = 0.03$, $P = 0.1$).

We then estimated the proportion of all observations in the BC-PCA assay that could be explained by the underlying transcriptome response using our model. A direct use of the $r = 0.46$ correlation (explaining ~21% of linear variance) may substantially underestimate the true correspondence between actual binary complex levels and transcription-based prediction of binary complex levels, given that even a perfect correlation would be degraded by unavoidable experimental measurement error. We therefore simulated a scenario in which mRNA expression explains all binary protein complex abundance changes, but is subject to the observed experimental variability in mRNA and BC-PCA measurements. In this model, expression levels are subject to the same uncertainty as observed between the 1 and 4 h time points and the resulting modeled BC-PCA ratios are subject to the observed uncertainty introduced by different biological replicates and DNA barcodes in ethanol (see Materials and Methods). Taking into account experimental error using this approach, we estimate that ~34% of the linear variance in the BC-PCA assay could be explained by transcriptome changes (Fig EV4D). Alternatively, when we related only the significant binary complex changes under ethanol to the mass action model, we estimated that 47% of significant changes correspond to the underlying transcriptome response (Fig EV4E). Because of the many approximations used in predicting binary complex level changes,

---

**Figure 4. Relating mRNA expression changes to PPI changes.**

A  Protein complex changes under the respiratory growth condition (i.e., in ethanol-containing medium) measured by BC-PCA (*y*-axis), compared to those predicted by a mass action model based on mRNA expression data (*x*-axis). Each dot represents a binary protein complex.

B  Percent of mRNA-based predictions validated by BC-PCA as a function of the predicted effect size (measured as $\log_2(R)$, averaged between UP- and DOWN-tags). As the mRNA-predicted effect size increases, the rate of validation increases for both accumulated and diminished PPIs. Lightly shaded areas show the 95% confidence intervals determined by 1,000 iterations of data resampling.

C  First-neighbor dynamic interactions of hexose transporter protein 1 (Hxt1), and the heat-shock protein 30 (Hsp30). Nodes are colored according to their transcriptional change observed at 4 h after the shift from glucose to ethanol-containing media. Edges are colored according to the complex abundance change, either predicted by mRNA changes (left) or observed by BC-PCA (right).

D  First-neighbor dynamic interactions of Lsp1, using the same schematic as in (C).

E  mRNA-based prediction accuracy for complex abundance changes, comparing binary complexes involving hubs having a directional bias in ethanol (see Fig 3B) with binary complexes involving other hubs. Error bars show 95% confidence intervals in the accuracy.

F  Density histogram of protein complex log-ratios (*x*-axis) resulting from selective repression of *RBD2*. The log-ratios of first- and second-neighbor Rbd2 interactions (i.e., complexes that contain Rbd2-interacting proteins, but not Rbd2 itself) are shown in purple and blue, respectively. All other interactions are shown in gray.

we take both these figures as a conservatively low estimate of the contribution of mRNA to the observed binary complex level changes.

We next sought to determine qualitative accuracy of an mRNA-based protein–protein interaction predictor, in which binary complexes are more simply predicted to either increase or decrease in level under a given environmental change. Using a stringent filter for mass-action-predicted binary complex abundance change ($|\log_2(R)|$ 2), 75% of predictions of accumulated complexes and 86% of depleted complexes were verified by the BC-PCA assay

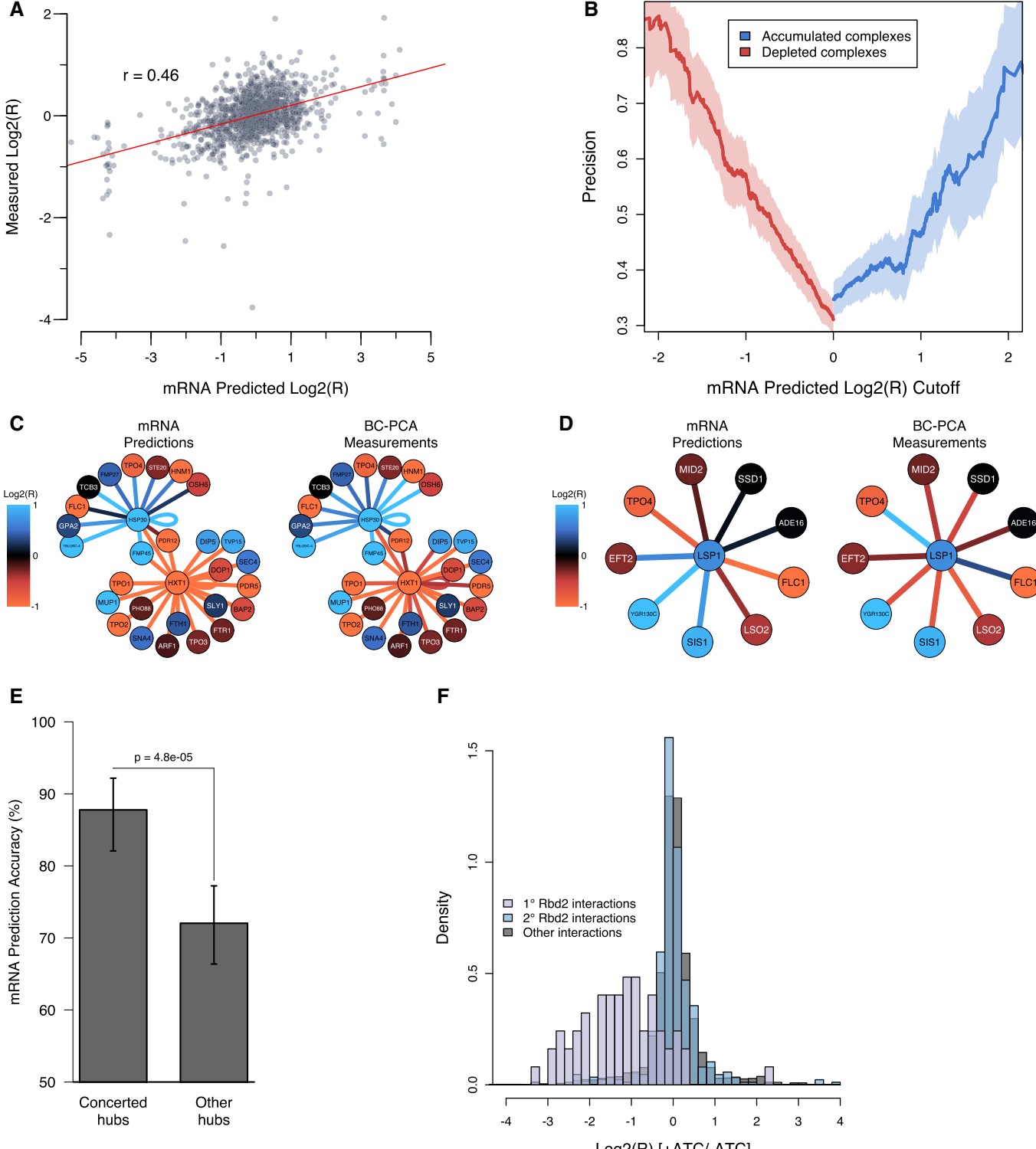

**Figure 4.**

(Fig 4B) indicating that the use of mRNA levels can predict accumulated or depleted binary complexes with high specificity during a diauxic shift.

The success of our mRNA-level-based predictions is consistent with protein-centric changes being responsible for many binary complex abundance changes. For example, the mass action model accurately predicted that numerous binary complexes involving the glucose transporter Hxt1 become depleted during respiration, based on Hxt1 transcript levels that were significantly reduced in this condition (Fig 4C). Similarly, the increase in *HSP30* transcript abundance led to the mass-action-based model predicting accumulated complexes consistent with our observed data (Fig 4C). These results suggest that the transcriptional regulation of these hubs during the shift from fermentative to respiratory growth influences a large number of protein interaction network changes.

We also found binary complex abundance changes which appear to be unrelated to an underlying transcriptome response. For example, Lsp1, a cortical patch protein associated with endocytosis, is a hub in which many interactions were changed that did not correspond to mRNA-based predictions (Fig 4D). For these binary complexes, many alternative modes of regulation may be possible, such as changes in protein stability, localization, or post-translational modification (e.g., phosphorylation, Mascaraque *et al*, 2013), although these mechanisms were not investigated here. Given the observation of many concerted hubs in ethanol and that Lsp1 was not observed to exhibit concerted behavior (Fig 3B), we sought to determine whether "concerted" hubs were more consistent with the underlying transcriptome response than other hubs. We found that mRNA-based predictions were 88% accurate when predicting the direction of significant binary complex abundance changes involving concerted hubs, compared to 72% when predicting binary complex abundance changes involving other hubs (Fig 4E, $P = 4.8e\text{-}05$). Thus, transcriptional change response is a useful predictor for all hubs, but more accurate for hubs with "concerted" binary complex abundance changes. This suggests that transcription is a predominant cause of the concerted hub changes in ethanol and that interaction-specific changes around other hubs likely diminish its predictive power.

### Transcriptional repression of the hub Rbd2 directly affects the abundance of binary protein complexes

To verify an *in vivo* causal relationship between mRNA expression and binary protein complex abundance in at least one example, we implemented a Tet-inducible CRISPRi system to repress the transcription of a target gene in a controlled manner. In this system, anhydrotetracycline (ATc)-induced expression of a guide RNA (gRNA) recruits a dCas9-Mxi1-repressor fusion protein to a locus of interest (Smith *et al*, 2016, 2017). We attempted to repress several "hub" genes by creating inducible gRNAs targeting their promoters, and validation by quantitative PCR showed that the gRNA targeting the *RBD2* gene displayed the clearest effects (Fig EV5A), with *RBD2* transcripts repressed ~8-fold (Fig EV5B). The BC-PCA pool was therefore transformed with a plasmid encoding the *RBD2*-gRNA and dCas9-Mxi1 fusion protein and then grown in selective media in the presence or absence of ATc (see Dataset EV2). Collectively, the growth rates of the 32 strains representing Rbd2 binary complexes were significantly reduced compared to other strains in the pool

(Fig 4F, $P = 2.8e\text{-}13$), and thus reducing the transcript levels of Rbd2 had clear effects on the abundance of Rbd2-containing binary complexes. In principle, altering transcript levels could lead to a cascade of perturbations in the PPI networks beyond a protein's first neighbors. However, we found no evidence of this phenomenon for Rbd2's second neighbors (Fig 4F, $P = 0.41$), and therefore, our results are consistent with the predicted effects of concentration changes remaining largely confined to the local network (Maslov & Ispolatov, 2007).

## Discussion

Here, we described the multiplex *in vivo* measurement of 1,379 protein–protein interactions in 14 environmental conditions, to our knowledge the most extensive direct study of how protein interaction networks respond dynamically to extrinsic environmental perturbations. The most striking finding was the prevalence of dynamic binary complexes. More than half of the PPIs we considered (757 of 1,379) responded to at least one perturbation. The environmental perturbations that yielded the largest number of changes relative to our reference condition were respiratory growth in ethanol, heat shock, oxidative stress, and DNA damage. That these responses were the most profound might have been expected, as these conditions are likely to have been frequently experienced in the evolutionary history of yeast (Gasch & Werner-Washburne, 2002; Gasch, 2007), allowing for selection and maintenance of a complex adaptive regulatory strategy. We observed that proteins with certain functions were more likely to participate in dynamic binary complexes. For example, proteins localized to the plasma membrane were enriched for participation in dynamic binary complexes, consistent with known plasma membrane protein changes in response to salt stress (Szopinska *et al*, 2011), and more generally to the known role of transmembrane gene families such as the ABC transporters in the stress response (Jungwirth & Kuchler, 2006). Although some dynamics were to be expected, we were surprised by the widespread nature of the phenomenon.

Functional analysis of PPIs that changed in specific conditions revealed both known and novel relationships. For example, the immunosuppressant FK506 enhanced the interaction between drug-efflux pumps (such as Pdr5 and Snq2) as previously described (Schlecht *et al*, 2012). Here, we found that binary complexes involving Yor1 (a multidrug transporter homologous to the human cystic fibrosis transmembrane receptor) also accumulated in FK506. Among the novel chemical-dependent changes we identified here was the depletion of binary complexes involved in yeast's lipid metabolic process by atorvastatin, an inhibitor of HMG-CoA reductase and a widely used cholesterol-lowering drug (Endo, 1992). In some cases, the changed interactions had a clear relationship to the environmental perturbation. For example, growth in methionine-supplemented media led to depletion of binary complexes containing the methionine permease Mup1, as well as other proteins important for methionine metabolism.

The patterns of PPI changes we observed, such as concerted "hub-centered" binary complex abundance changes, were consistent with a model wherein numerous PPI changes occur as a result of alterations in a single or small number of proteins. This suggests that control at the protein level (as opposed to the interaction level)

is particularly important for understanding PPI network rewiring. Future analyses might yield an expanded view of dynamic PPI modularity. For example, using co-expression data, hub proteins have been divided into "party" and "date" hubs based on the degree of co-expression with their partners (Han *et al*, 2004). Party hubs are predicted to interact with all their partners at the same time and space, whereas date hubs are predicted to interact with their partners at different times and/or spaces. More extensive dynamic interaction analysis may reveal distinct roles for these two different hub types in PPI modulation.

We found a striking correlation between transcript abundance and binary protein complex abundance changes during a carbon source shift from glucose to ethanol, which we estimate to explain ~47% of the observed changes. This observation systematically extends the small number of examples of expression-dependent interactions previously described in other environments (Schlecht *et al*, 2012). While we found a strong concordance between mRNA expression and binary complex levels during a diauxic shift, this relationship may vary depending on the condition. Changes in protein localization may have the capacity to impact complex levels more than changes in total protein abundance (Levy *et al*, 2014). Indeed, Rochette *et al* (2014) observed that the effects of methyl methanesulfonate (MMS) on differential localization explained more modulated PPIs than did differential protein abundance, although both had significant explanatory power. The recent availability of high-throughput conditional protein localization and abundance data (Chong *et al*, 2015) should allow a straightforward extension of a mass-action-based model (guided by BC-PCA experiments in the same conditions) that accounts for changes in co-localization (as well as expression) to explain dynamic protein complexes.

Identifying cases where transcription does not play a direct role may be of special interest in investigating other mechanisms of binary complex abundance changes. For example, we found that mRNA levels in ethanol were not predictive of binary complex abundance changes involving Lsp1 (Fig 4D). This assay could be used to prioritize investigations of post-translational modification [such as phosphorylation in the case of Lsp1 (Mascaraque *et al*, 2013)] on the observed binary protein complex abundance changes.

One strong feature of our assay is the small culture volume (700 μl), which enables interrogation of a large number of conditions (including small molecules that are available only in limited quantities) in parallel. The assay is also readily amenable to next-generation sequencing approaches that quantify molecular barcode sequence abundance (Smith *et al*, 2009, 2010), which could further enhance assay throughput and resolution. More broadly, the pooled format makes it straightforward to perform direct manipulations and investigate the resulting PPI changes. For example, we were able to directly relate transcription changes to PPI changes using a CRISPRi-based approach and found that the down-regulation of the *RBD2* gene led to a specific loss of many direct interactions of its gene product. Combining BC-PCA with targeted transcriptional repression could be used to systematically validate quantitative predictions of transcriptionally mediated network effects. Such an approach could be used, for example, to study the architecture of large complexes by altering constituent protein levels in a controlled manner (Diss *et al*, 2013), in understanding gene knockdown and overexpression phenotypes, and ultimately to predict downstream functional consequences of altered gene expression.

A limitation of our assay is that we are restricted to a set of PPIs known in a "standard" reference condition. While our assay is able to determine both accumulation and depletion of interactions within this standard set, condition-specific interactions where the interaction is undetectable in the reference were beyond the scope of this study. The large number of dynamic binary complexes we identified suggests that condition-specific interactions are prevalent. Finding such interactions will require further improvements in the scalability of PCA, as performing a separate exhaustive screen in every condition tested using the current approach is not feasible. Another limitation is that the mDHFR PCA uses survival and growth as a readout of PPIs. Thus, while it can be used to understand overall binary protein complex abundance changes, it may not capture highly transient dynamic interactions. Applying BC-PCA with variants based on reversible fluorescence and performing selection using cell sorting may enable multiplexed analysis of transient interaction changes (Remy & Michnick, 1999; Morell *et al*, 2008; Li *et al*, 2014; Tchekanda *et al*, 2014).

The *in vivo* nature of BC-PCA should readily allow future integration of genome-scale phenotype data to global protein–protein interaction remodeling. The apparent prevalence of "protein-centric" changes strongly warrants the investigation of other phenomena such as differential localization and post-translational modification. Ultimately, this may guide the understanding of *in vivo* PPI remodeling principles.

# Materials and Methods

### Media and growth conditions

Rich media (YPD) consisted of 1% yeast extract, 2% bactopeptone, and 2% glucose. Minimal media contained 1.7 g yeast nitrogen base, 20 g dextrose, 5 g ammonium sulfate, 50 mg histidine, 50 mg leucine, 20 mg uracil per liter (non-selective media). This media was supplemented with 100 μg/ml methotrexate (MTX) to select for binary protein complexes (selective media). Isogenic cultures (100 μl) were inoculated at a concentration of 0.02 $OD_{600}$/ml and grown in 96-well microtiter plates at 30°C. Optical density was measured every 15 min over the course of several hours (as indicated in graphs) using a GENios microplate reader (Tecan). The growth rate of a strain was calculated as follows: (i) the first 10 OD readings were averaged and subtracted from all OD readings of the corresponding curve in order to set the baseline of the growth curve to zero, (ii) the area under the curve (AUC) was then calculated as the sum of all OD readings. 300 reads (corresponding to 75 h) were used to verify previously published PCA strains in Fig EV1. In Fig EV2, where we compared the growth rate of each strain in the presence of condition to a control, a relative growth value was calculated as follows: $(AUC_{condition} - AUC_{control})/AUC_{control}$.

### Chemical reagents and environmental conditions

FK506 (catalog no. 10007965) was purchased from Cayman Chemical. Methotrexate (catalog no. M4010), doxorubicin (catalog no. D1515), D-sorbitol (catalog no. S1876), L-methionine (catalog no. M-9625), copper(II) sulfate (catalog no. 451657), sodium chloride (catalog no. S7653), anhydrotetracycline hydrochloride (catalog no.

37919), hydrogen peroxide solution (catalog no. 216763), atorvastatin calcium salt trihydrate (catalog no. PZ0001) were purchased from Sigma-Aldrich. A full description of environmental conditions is listed in Dataset EV1.

### PCA pool construction and microarray experiments

We purchased the Yeast Interactome Collection (YSC5849) from Open Biosystems and used it as a source of mDHFR-tagged strains. We chose 2,394 of the 2,770 PCA-PPIs published in (Tarassov *et al*, 2008) and cherry-picked the corresponding *MAT*a (mDHFR-F[1,2]-NatMX fusions) and *MAT*α (mDHFR-F[3]-HphMX fusions) strains. The *MAT*a strains were mated individually with their *MAT*α partner strains on high-density arrays using a Singer ROTOR robot. Diploids were selected on rich media supplemented with hygromycin B and nourseothricin. To determine the growth rate of each individual strain, we monitored optical density of isogenic cultures in selective media for 75 h using a GENios microplate reader (Tecan). The growth rate was calculated as the area under the curve (AUC) as described above. Strains with AUC > 8 were defined as hosting a binary protein complex; 1,701 strains passed this step. The corresponding 1,701 *MAT*a strains, as well as several additional strains of interest, were then tagged with unique barcodes by integrative transformation into the HO-locus. A small number of PPIs were also tagged with two different barcodes. Barcodes were constructed as described previously (Schlecht *et al*, 2012). Tagged strains were then mated individually with their *MAT*α partner strains on high-density arrays using a Singer ROTOR robot, and diploids were selected on minimal media lacking methionine and lysine. 1,432 strains, representing 1,428 unique interactions, were successfully transformed and mated. To construct the pool of BC-PCA strains, all colonies were scraped off plates and resuspended in YPD. Aliquots of the pool were kept as frozen stocks ($OD_{600}$/ml = 9). Frozen aliquots of the pool were recovered for 10 generations in non-selective media, and then cells were diluted in selective and non-selective media to an $OD_{600}$/ml of 0.02. Cells were then grown in the presence of DMSO (1%), a compound, or a specific condition (a panel of 14 perturbations in total) in 700 μl cultures in a 48-well microplate. After three pool doublings, cells were harvested and pellets stored at −20°C. Genomic DNA was extracted with the YeaStar Genomic DNA kit (Zymo Research, catalog no. D2002) and using the QIAXtractor (Qiagen). PCR-amplification of barcodes and hybridization to Genflex tag 16k arrays (Affymetrix) were performed as described previously (Suresh *et al*, 2016).

### Identification of dynamic binary protein complexes

All cells carry two tags that hybridize to the Genflex tag 16k array, an "UP-tag" and a "DOWN-tag". From each array, we extracted the fluorescence intensity values for every UP-tag and DOWN-tag associated with the 1,432 strains in the barcoded PCA pool. These raw fluorescence values were quantile-normalized and then $log_2$-transformed using the normalize.quantiles function of the preprocessCore package in R. Quantile normalization was performed separately for UP-tags and DOWN-tags, as these samples were PCR-amplified separately and therefore may contain a different amount of labeled nucleic acid material (Suresh *et al*, 2016). Fourteen experimental conditions were measured in duplicate, and the control condition (DMSO) was measured with six replicates. To calculate a $log_2$-ratio between the control and experimental condition ($log_2(R)$), the mean of the six control replicates was subtracted from the mean of the two treatment replicates. This $log_2$-ratio was calculated separately for UP- and DOWN-tags. A moderated *t*-statistic was then calculated using the R package LIMMA, and the derived *P*-values were further converted to *q*-values using Benjamini and Hochberg false discovery rate (FDR) correction using the *P*.adjust function in R. A separate *q*-value was calculated for UP- and DOWN-tags.

PCA strains with a significant increase in their tag abundance in response to an experimental condition were identified using *q*-value < 0.05 (for both UP-tag and DOWN-tag) and $log_2(R) > 0.25$ (for both UP- and DOWN-tag) as cutoffs. PCA strains with a significant decrease in tag abundance were identified using *q*-value < 0.05 (for both UP-tag and DOWN-tag) and $log_2(R) < -0.25$ (for both UP- and DOWN-tag) as cutoffs. The complete list of *q*-values and *R* values for all experimental conditions is provided in Dataset EV2. CEL files are available via the NCBI's Gene Expression Omnibus under accession number GSE72425.

### Identification of PCA-fragment tag effects

In order to systematically account for cases where the presence of the mDHFR-fragment interferes with the function of the tagged protein, we included control cultures grown in the same 14 environmental conditions but in non-selective media. These control experiments were also performed in duplicate. To identify problematic strains, we chose the same cutoffs for log-ratios ($log_2(R) > 0.25$ or $log_2(R) < -0.25$ for both UP- and DOWN-tags, in the same direction) and statistical significance (*q*-values < 0.05 for both UP- and DOWN-tags) as above. We identified several strains meeting these criteria, indicating that in these strains, the protein tags had heterozygous condition-dependent effects on growth. In most conditions, the number of such cases was small (i.e., < 2% of all strains in the pool were affected); however, growth in ethanol as an alternative carbon source and the exposure to hydrogen peroxide identified a larger number of strains with significant changes in growth rate in non-selective media. In most cases, the introduction of the PCA fragment led to a decrease in growth rate. From each condition, we excluded from further analysis all complexes that were found to elicit a condition-specific fitness effect in the absence of methotrexate selection (Dataset EV2). In total, 176 of 1,383 strains displayed a selection-independent growth change in at least one condition (Dataset EV2). In addition, we also excluded all binary protein complexes containing a specific F[1,2]- or F[3]-fragment if that fragment occurred more than once in the list of strains that displayed a selection-independent growth change. For example, 16 of the 25 strains containing HNM1-F[1,2] showed a significant growth defect in ethanol without the presence of MTX. Therefore, all 25 HNM1-F[1,2] containing strains were excluded from the ethanol dataset from further analysis. These strains are listed in Dataset EV2 (in the excluded –MTX tab) and are marked as "by association".

### Hierarchical clustering

For the cluster analysis in Fig 1A, we included all 757 complexes which displayed a significant change in complex abundance (i.e.,

$\log_2(R) > 0.25$ or $\log_2(R) < -0.25$ for both UP- and DOWN-tags, and $q$-value $< 0.05$ for both UP- and DOWN-tags) in at least one of the conditions. Clustering along both experiment and PPI axes was performed on $\log_2(R)$ values (averaged UP- and DOWN-tags) using Gene Cluster 3.0 and Java Treeview. For hierarchical clustering, experiment and gene data were first median-centered and then clustered using the Correlation (uncentered) similarity metric and average linkage clustering.

### GO term enrichment analysis

To analyze GO term enrichment among frequently perturbed interactions (changing in four or more conditions), we took the set of genes connected to complexes which participated in these interactions, and used the set of all genes in the pool as the background set.

For GO analysis in each condition, we took the set of genes connected to complexes which had changed interaction strength in a desired direction in a given condition (either accumulated or depleted) and then ranked the genes by the effect size (maximum $|\log_2(R_{DOWNtag}) + \log_2(R_{UPtag})|$ of its significantly changed interactions in the desired direction) in decreasing order, with ties shuffled. The background was taken as the set of genes participating in any non-excluded interaction measured in that condition.

All GO enrichment analyses were performed using the FuncAssociate 2.1 web service (Berriz *et al*, 2003; Berriz *et al*, 2009), which takes as input list of genes (ranked or unranked) and calculates significance of enrichment for each GO term by comparing the observed nominal $P$-value from a hypergeometric test for that GO term with the most significant nominal $P$-value observed from any GO term using either random collections of genes (when performing an unranked test) or randomly ranked lists of similar size (when performing a ranked test), drawn from the same background set of genes. The following parameters were used in the FuncAssociate 2.1 webserver: The GO association file was taken from the provided *Saccharomyces cerevisiae* annotations in the sgd_systematic namespace with default evidence codes (EXP, IDA, IPI, IMP, IGI, IEP, ISS, ISO, ISA, ISM, IGC, RCA, TAS, NAS, IC, IEA). For each query, 10,000 iterations were used, with an adjusted significance cutoff of $P\_adj$ ($q$) $< 0.05$. The search was only performed for over-represented terms.

### Dynamic subnetwork simulations

In Fig 3C, we generated random dynamic subnetworks for each condition by sampling from the set of all measured BC-PCA interactions in that condition with different sampling procedures that yielded varying proportions of protein-centric and interaction-specific effects and examined which procedure yielded random dynamic subnetworks that were the most consistent with the largest component sizes observed in our data. To this end, we performed iterative random sampling in which the number of accumulated, depleted, and unchanged interactions observed in a certain condition were generated using a combination of two models: protein centric (referred here as "node-based") and interaction specific (referred here as "edge-based"). In each scenario, a given number of interactions were first sampled using the node-based method, and the remainder were then sampled using the edge-based method.

We used a node-based sampling procedure in which a random protein in the network (i.e., a node) is first proposed to have all of its interactions change in a given direction. If these proposed changes would not cause the simulation to exceed the desired number of node-based dynamic interactions in that direction, they are accepted and the dynamic subnetwork is modified. Otherwise, another node is chosen at random until this condition is satisfied. After successfully sampling one node, the simulated direction is switched (e.g., from "accumulated" to "depleted") and this process is repeated until the desired number of both accumulated and depleted interactions are sampled. If the limit of dynamic interactions in a given direction has been reached, further sampling steps in that direction are skipped. If at any point in this sampling process there is a conflict in the assigned direction (i.e., an interaction was previously assigned to change in one direction but the current sampling step assigns the opposite direction), then the most recent assignment takes precedence.

For "edge-based" sampling methods (referred to as "interaction-specific" sampling in the text), the desired number of accumulated and depleted interaction labels is assigned uniformly at random to existing edges. In scenarios involving a combination of the two models, "edge-based" interactions are randomly assigned to the list of static edges in the network (i.e., those which were not already sampled using the "node-based" method).

After successfully simulating a dynamic subnetwork using the desired sampling method(s), the largest connected component (defined as the number of nodes in the largest subgraph such that any two proteins in that subgraph can be connected by dynamic interactions of the same sign) was determined using the igraph package in R for both the accumulated and depleted dynamic subnetworks. The size of the largest connected component ("largest component size") in each of 1,000 simulated networks was determined separately for accumulated and depleted interactions. From simulations, we extracted the 5 percentile and 95 percentile values for both accumulated and depleted subnetworks. This allowed us to indicate cases where an observed network was consistent with simulations; that is, they had largest component sizes that fell between these values for both accumulated and depleted subnetworks (Fig 3C, gray shading). Cases where the observed network behavior was inconsistent are indicated (Fig 3C; orange if either observed largest component exceeds 95 percentile or blue if either observed largest component falls below 5 percentile). For both accumulated and depleted interactions, departure of a simulation from observation was measured by difference between the percentile of the observed largest component size and 50 percentile. The most consistent simulation overall was taken to be that which minimized the largest departure observed for accumulated and depleted interactions (Fig 3C). The same analysis was repeated for Fig 3D using overall graph density rather than component size as a comparison metric to the BC-PCA data. Graph density was measured as $2|E|/|V|(|V|-1)$ where $|E|$ is the number of complex changes in a given direction and $|V|$ is the number of nodes participating in at least one complex change in that direction.

### Analysis of binary protein complexes containing hub proteins

For the analysis depicted in Fig 3B, we first defined 74 "hubs" as proteins having 10 or more interaction partners in the network.

Binary complexes containing a hub protein were then identified. In order to systematically define "directional bias", we first extracted the $\log_2(R)$ values (for both UP- and DOWN-tag) measured in the 14 conditions for binary complexes involving these hubs. For each UP- and DOWN-tag pair corresponding to a given interaction, we used the $\log_2(R)$ value which was closest to zero as a conservative measure of change in binary complex level. For each hub in each condition, we then compared the conservative $\log_2(R)$ values associated with that hub to all other conservative $\log_2(R)$ values. For example, in the case of Hxt1, a protein that is present in 23 interactions in the network (represented by 23 strains), the conservative $\log_2(R)$ values of these 23 strains were compared to the conservative $\log_2(R)$ values of the remaining 1,360 strains in each condition. A Mann–Whitney *U*-test was performed comparing these two $\log_2(R)$-value distributions. The derived *P*-values were corrected for multiple testing using the *P*.adjust function in R with the BH correction method. Figure 3B plots the directional bias of 50 hubs showing bias in at least one condition ($q < 0.05$ for both "complex depletion bias" and "complex accumulation bias").

**Quantitative RT-PCR experiments**

RNA was extracted from BC-PCA pool strains transformed with CRISPRi plasmids targeting the RBD2 locus. An overnight culture of the pool was diluted to an $OD_{600}$/ml of 0.03 and grown for 24 h in minimal media supplemented with leucine and histidine and in the presence or absence of 250 ng/ml anhydrotetracycline (ATc). Cultures (5 ml) were then harvested, and total RNA was extracted using the RiboPure-Yeast kit (Ambion, catalog no. AM1926). cDNA was synthesized in 20 μl reactions using the High-Capacity RNA-to-cDNA Kit (Applied Biosystems, catalog no. 4387406) and then diluted 10-fold using nuclease-free water. Primers for quantitative PCR were designed using primer3; the program's default settings were used to select primers that produced 75- to 125-bp amplicons. Quantitative PCR mixes (10 μl) contained 1× SYBR Green PCR Master Mix (Applied Biosystems, catalog no. 4309159) and 5 μM of each primer. Data were collected and analyzed on a StepOne Real-Time PCR System (Applied Biosystems). Ct-values report the cycle at which SYBR fluorescence crosses a threshold; the threshold was automatically set at a point within the exponential phase of the PCRs. Gene-specific differences in $C_t$-values were calculated by subtracting $C_t[\text{ATc;gene}] - C_t[\text{DMSO;gene}]$ ($\Delta C_t[\text{ATc;gene}]$) and normalizing against the $\Delta C_t$ for the control gene *ACT1* ($\Delta C_t[\text{ATc;ACT1}]$). Ratios reported were calculated from this equation:

$$2^{((\Delta C_t[\text{ATc;ACT1}]) - (\Delta C_t[\text{ATc;gene}]))}.$$

**Total RNA isolation, cDNA target synthesis, and GeneChip hybridization**

For the expression time course experiment described in Fig 4, a PCA pool culture was grown overnight in non-selective media and then resuspended in selective media containing ethanol instead of dextrose as the sole carbon source. 10 ml aliquots were then harvested at 0, 0.5, 4, and 12 h, and total RNA was extracted from these samples using the RiboPure-Yeast kit (Ambion, catalog no. AM1926). cDNA was synthesized in 10 μl reactions containing

1 μg/μl total RNA, 12.5 ng/μl Oligo(dT)12–18 primer (Invitrogen, catalog no. 18418-012), 15 units/μl SuperScript II (Invitrogen, catalog no. 18064-014), 1× First Strand Buffer, 10 mM DTT, and 10 mM dNTPs (Invitrogen, catalog no. 18427013). After the RNA and primers were denatured for 10 min at 70°C, the remaining reagents were added, and the reaction was incubated at 42°C for 60 min. To remove the RNA template, two units of RNase H were then added and the mix was incubated at 37°C for 20 min and then at 95°C for 5 min. Quality of total RNA and cDNA was monitored using RNA Nano 6000 chips processed using the 2100 BioAnalyzer (Agilent). 220 μl hybridization cocktail containing heat-fragmented and biotin-labeled cDNA at a concentration of 0.05 μg/μl were injected into GeneChips and incubated at 45°C on a rotator in a Hybridization Oven 640 (Affymetrix) overnight at 60 rpm. The arrays were washed and stained with a streptavidin–phycoerythrin conjugate (SAPE; Molecular Probes). The Gene Chips were processed in a GeneArray Scanner (Agilent) using the default settings. CEL files containing the raw data were computed from DAT array image files using the statistical algorithm implemented in MAS 5.0 (Affymetrix). Log2-transformed raw data were preprocessed (background adjustment, normalization, and summarization of probe sets) by using the Robust Multiarray Analysis (RMA) package from BioConductor. CEL feature-level data are available via the NCBI's Gene Expression Omnibus under accession number GSE72425.

**Predicting protein complex levels and complex level abundance changes using gene and protein expression**

In order to predict the change in each protein complex using gene and protein expression, we modeled every complex as binary and acting independently. We took the resting total cellular concentration of two proteins ($C_{P1}, C_{P2}$) in a complex as the value reported in PaxDB (Wang *et al*, 2012; values given are in ppm). When no concentration was available, the median genomewide value was used instead. Using a previously established approach (Maslov & Ispolatov, 2007), we modeled the dissociation constant ($K_d$) of each complex to be proportional to the maximum total concentration of the proteins in the complex $\max(C_{P1}, C_{P2})/20$. Using the law of mass action $K_d = \frac{(C_{P1} - C_{P1::P2})(C_{P2} - C_{P1::P2})}{C_{P1::P2}}$, an expression can be made in terms of the predicted complex concentration $C_{P1::P2}$:

$$C_{P1::P2} = \frac{1}{2}\left(-\sqrt{C_{P1}^2 - 2C_{P1}(C_{P2} - K_d) + (C_{P2} + K_d)^2} + C_{P1} + C_{P2} + K_d\right).$$

In the limit where $K_d$ approaches 0, this reduces to:

$$\lim_{K_d \to 0} C_{P1::P2} = \frac{1}{2}\left(-\sqrt{C_{P1}^2 - 2C_{P1}C_{P2} + C_{P2}^2} + C_{P1} + C_{P2}\right)$$
$$= \frac{1}{2}\left(-\sqrt{(C_{P1} - C_{P2})^2} + C_{P1} + C_{P2}\right)$$
$$= \frac{1}{2}(C_{P1} + C_{P2} - |C_{P1} - C_{P2}|)$$
$$= \min(C_{P1}, C_{P2}).$$

In Fig EV1C, $\min(C_{P1}, C_{P2})$ was used to approximate an estimated protein complex level for direct comparison to Freschi *et al* (2013). In Figs 4 and EV4, the full formula for $C_{P1::P2}$ was used to

estimate $C_{P1::P2_{\text{EtOH}}}/C_{P1::P2_{\text{DMSO}}}$ as the predicted ratio of each complex based on the mRNA expression data. To estimate $C_{P1::P2_{\text{EtOH}}}$, we first estimated a concentration ratio ($R$) of each protein under a diauxic shift ("EtOH") from the mRNA expression data according to the change in processed fluorescence intensity ($I$) from the control condition ("DMSO"), such that $R_{P1} = I_{\text{mRNA1}_{\text{EtOH}}}/I_{\text{mRNA1}_{\text{DMSO}}}$ and $R_{P2} = I_{\text{mRNA2}_{\text{EtOH}}}/I_{\text{mRNA2}_{\text{DMSO}}}$. $R$ was assigned as 1 for proteins with missing mRNA expression measurements in either condition. Then, $R_{P1}C_{P1}$ was assigned as $C_{P1_{\text{EtOH}}}$ and $R_{P2}C_{P2}$ was assigned as $C_{P2_{\text{EtOH}}}$ to calculate $C_{P1::P2_{\text{EtOH}}}$. Strains with a lack of barcode correspondence ($|\log_2(R_{\text{UPtag}}) - \log_2(R_{\text{DOWNtag}})| > 1$) were excluded in Figs 4 and EV4.

## Precision of protein complex abundance estimates from our mass action model

Even perfectly accurate and precise predictions of protein complex abundance will exhibit imperfect correlation with observation, because of experimental measurement error in both expression and BC-PCA data. To isolate errors intrinsic to the model from those caused by experimental variation, we carried out a multistep analysis. First, we estimated experimental error in both expression and BC-PCA data. Second, we used a generative model to produce "noise-added" BC-PCA data by: (i) adding experimental error to the observed expression data to produce replicates of simulated expression data; (ii) using our mass action model to generate a predicted BC-PCA dataset from each simulated expression dataset; and (iii) adding experimental error to each replicate predicted BC-PCA dataset. Correlation was then measured between "noise-added" PCA replicates. Separately, we used the expression data more directly to predict BC-PCA data from our mass action model and measured correlation between predicted and observed BC-PCA data. Disagreement between prediction and observation stems from model inaccuracy as well as errors in expression and BC-PCA data. By contrast, disagreement between "noise-added" BC-PCA datasets depends only on errors in expression and BC-PCA data. Thus, the extent to which correlation between prediction and observation is lower than correlation between "noise-added" replicate BC-PCA data allowed us to estimate intrinsic model error.

In order to create "noise-added" BC-PCA data, the first goal was to figure out how much noise to add in steps (i) and (iii) so that it reasonably reflects the uncertainty in the experiment. Intuitively, the experimental noise at each step can be estimated using the correlation between two replicate observations. If we treat the correlation between two replicate observations as reflecting the correspondence between two noisy measurements of an "ideal" variable $X$ with random error $\varepsilon_1$ and $\varepsilon_2$, then we can derive the amount of error to add to X in order to achieve the observed correlation $\rho(X + \varepsilon_1, X + \varepsilon_2)$. The correlation between two measurements of an "ideal" variable $X$ with random errors $\varepsilon_1$ and $\varepsilon_2$ such that $\sigma_{\varepsilon_1} = \sigma_{\varepsilon_2}$ is

$$\rho(X + \varepsilon_1, X + \varepsilon_2) = \frac{\text{cov}(X + \varepsilon_1, X + \varepsilon_2)}{\sigma_{X+\varepsilon_1}\sigma_{X+\varepsilon_2}} = \frac{\sigma^2 X}{\sigma^2 X + \sigma^2_{\varepsilon_1}}.$$

Thus, the amount of normally distributed error to add to "reference" experimental data in our simulation such that two noisy observations are expected to have the same correlation as seen in the experiment is:

$$\sigma_{\varepsilon_1} = \sigma_X \sqrt{\frac{1}{\rho(X + \varepsilon_1, X + \varepsilon_2)} - 1}.$$

In step (i), the simulation takes gene expression ratio as input for each protein complex pair. We used the $\log_2$-transformed expression ratio between $t = 0$ h and $t = 4$ h under ethanol as the reference ($X$) to which noise was added. In order to estimate the amount of noise to add to this input using the above formula, we used the correlation between the $t = 1$ h and $t = 4$ h ratios, as these observations had the most correspondence to each other. In step (ii), the model then takes these "noisy" expression ratios (as well as an estimate of resting concentrations for each protein, to which we did not simulate any error) to give an estimated PCA output. In step (iii) we then take the output from this model and use the same approach as in (i) to add normally distributed error to the output of the model. In this case, however, the error reflects both the experimental error introduced by biological variability (by taking the observed correlation between the two replicates for the same DNA tag) and DNA tag variability (by taking the observed correlation within the same replicate, using different DNA tags). To determine the distribution of our estimates, we simulated 100 "noise-added" replicates and performed a pairwise correlation for all $\binom{100}{2} = 4,950$ combinations, each of which estimate $\rho^2_{(X,X+\varepsilon_{\text{exp}})}$.

Given a distribution of estimates for $\rho^2_{(X,X+\varepsilon_{\text{exp}})}$, our second goal was to estimate the "intrinsic" model correspondence to the data $\rho^2_{(X,X+\varepsilon_{\text{mod}})}$. For this estimate, we use the difference between the correlation of "noise-added" PCA datasets (which depends only on experimental errors) and a difference in correlation between the model predictions and observed experimental output (which depends both on experimental errors and intrinsic model errors). The correlation of a variable $X$ compared to the same variable with added random error ($X + \varepsilon$) can be expressed as $\rho^2_{(X,X+\varepsilon)} = \frac{\sigma^2 X}{\sigma^2 X + \sigma^2_\varepsilon}$. Using this definition, we can partition the total error variance $\sigma^2_\varepsilon$ into the variance introduced by experimental error ($\sigma^2_{\varepsilon_{\text{exp}}}$) and the residual variance, which we take in this case to be introduced by the "true" lack of model correspondence $\left(\sigma^2_{\varepsilon_{\text{mod}}}\right)$ : $\sigma^2_{\varepsilon_{\text{total}}} = \sigma^2_{\varepsilon_{\text{exp}}} + \sigma^2_{\varepsilon_{\text{mod}}}$, so that $\rho^2_{(X,X+\varepsilon_{\text{mod}})} = \frac{\sigma^2 X}{\sigma^2 X + \sigma^2_{\varepsilon_{\text{total}}} - \sigma^2_{\varepsilon_{\text{exp}}}}$. Using these relationships, it is possible to restate $\rho^2_{(X,X+\varepsilon_{\text{mod}})}$ as a comparison of the "noise-added" PCA correlation $\rho^2_{(X,X+\varepsilon_{\text{exp}})}$ and the overall model correspondence to the data $\rho^2_{(X,X+\varepsilon_{\text{exp}}+\varepsilon_{\text{mod}})}$:

$$\rho^2_{(X,X+\varepsilon_{\text{mod}})} = \frac{1}{\frac{1}{\rho^2_{(X,X+\varepsilon_{\text{exp}}+\varepsilon_{\text{mod}})}} - \frac{1}{\rho^2_{(X,X+\varepsilon_{\text{exp}})}} + 1}.$$

As we have obtained a distribution of $\rho^2_{(X,X+\varepsilon_{\text{exp}})}$ from "noise-added" replicates, we then sought to obtain a distribution of $\rho^2_{(X,X+\varepsilon_{\text{mod}})}$ estimates. To this end, we resampled the observed model-to-experiment correspondence to estimate a separate $\rho^2_{(X,X+\varepsilon_{\text{exp}}+\varepsilon_{\text{mod}})}$ for each of the 4,950 $\rho^2_{(X,X+\varepsilon_{\text{exp}})}$ estimates. We plotted the distribution of estimates generated by each of these $\rho^2_{(X,X+\varepsilon_{\text{exp}})}$ and $\rho^2_{(X,X+\varepsilon_{\text{exp}}+\varepsilon_{\text{mod}})}$ combinations in Fig EV4D and E.

## CRISPRi plasmid construction

We employed a single-plasmid system encoding the dCas9-Mxi1 repressor and guide RNA (gRNA) for inducible and targeted repression of the *RBD2* gene (Smith *et al*, 2016, 2017). The sequence

encoding the specificity-determining region of the *RBD2*-guide RNA is GAAGAATAGGGGGGATGGGAA.

## Availability of data and materials

All data used in the publication are available as EV files attached to the manuscript. CEL feature-level microarray data are available via the NCBI's Gene Expression Omnibus under accession number GSE72425. Scripts used for the analysis are available on GitHub: https://github.com/a3cel2/bc_pca_git.

## List of abbreviations

Protein–protein interaction (PPI), protein-fragment complementation assay (PCA), murine dihydrofolate reductase (mDHFR), methotrexate (MTX), base pairs (bp), area under the curve (AUC), dimethyl sulfoxide (DMSO), anhydrotetracycline (ATc), guide RNA (gRNA), clustered regularly interspaced palindromic repeats (CRISPR), CRISPR interference (CRISPRi).

**Expanded View** for this article is available online.

## Acknowledgements

The authors are grateful to Maureen Hillenmeyer and Sasha Levy for helpful discussions of the project. This work was supported by US National Institutes of Health grants P01HG000205 (R.W.D), HG005785 (R.W.D.), U01GM110706-02 (R.W.D.), and HG004233 (a "Centers of Excellence in Genomic Science" grant; F.P.R.) and by the Canada Excellence Research Chairs Program (F.P.R.) and Krembil Foundation (F.P.R.).

## Author contributions

US and RPS conceived the experiments. US, JS, SS, MM, AMA, MP, and RPS performed experiments. AC, US, WX, RPS and FPR analyzed the data. AC, US, RPS, and FPR wrote the paper. RWD provided valuable insight and advice.

## Conflict of interest

The authors declare that they have no conflict of interest.

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
