## [Review Process File · Molecular Systems Biology]

Quantitative analysis of protein interaction network dynamics in yeast

Albi Celaj, Ulrich Schlecht, Justin Smith, Weihong Xu, Sundari Suresh, Molly Miranda, Ana Maria Aparicio, Michael Proctor, Ronald W. Davis, Frederick P. Roth and Robert P. St.Onge

*Corresponding authors: Frederick P. Roth, University of Toronto
Robert P. St.Onge, Stanford University*

Review timeline:	Submission date:	10 January 2017
	Editorial Decision:	23 February 2017
	Revision received:	10 May 2017
	Editorial Decision:	12 June 2017
	Revision received:	13 June 2017
	Accepted:	14 June 2017

Editor: Maria Polychronidou

Transaction Report:

1st Editorial Decision

23 February 2017

Thank you again for submitting your work to Molecular Systems Biology. We have now heard back from the three referees who agreed to evaluate your study. As you will see below, the reviewers appreciate the broad scale at which condition-dependent protein-protein interactions are analyzed. However, they raise a number of concerns, which should be carefully addressed in a revision of the manuscript. The reviewers' recommendations are rather clear so I think that there is no need to repeat the points listed below.

REFeree REPORTS

Reviewer #1:

The manuscript, titled "Quantitative analysis of protein interaction network dynamics in yeast" by Schlecht et al., describes the establishment and measurement of changes in protein-protein interactions under a series of perturbations in the budding yeast. This is an expansion built upon several previously published studies. The authors implemented a highly multiplexed murine dihydrofolate reductase (mDHFR)-based protein complementation assay to quantitatively measure formation of binary protein-protein interactions in yeast. After introducing DNA barcodes to 1383 yeast strains that each harbored a binary PPI complex, which was fused to fragment of mDHFR, the authors pooled these strains and grew them together in the presence of methotrexate (MTX) under

14 different conditions. Because formation of a binary protein complex would reconstitute mDHFR, which renders the cells resistance to MTX in a quantity-dependent manner, they measured abundance of the DNA barcodes as a surrogate of quantitative measurement of binary protein complex formation. They found that many protein complexes were environment dependent. Interestingly, hub proteins (those with many known PPIs) showed concerted changes. To better understand the biology behind these observed changes in protein complexes, they integrated mRNA expression data into a diauxic shift from glucose to ethanol and observed that mRNA levels could predict concerted changes with 87% accuracy.

This is a very well designed large-scale study of PPI changes under 14 conditions built upon the success of several previous studies on a smaller scale. Compared with Y2H approach or mapping of protein complexes using IP-MS/MS, the presented approach enables examination of changes in binary PPI complexes under physiological conditions. I do not have any major concerns but only minor ones.

Minor concerns

1. Can the authors elaborate on their ability to detect relatively weaker PPIs?
2. Can different affinity of PPIs affect the abundance of the DNA barcodes?
3. Have the authors detected any kinase-substrate interactions in their assays?
4. On page 7 the authors stated that a significant correlation between growth rate and "pairwise protein abundance" was observed. Is this also true for those hub proteins?

Reviewer #2:

The manuscript describes an expansion of an existing methodology that they reported fairly recently in PNAS to monitor changes in a large number of protein-protein interactions following perturbations of cells. I thought that it was a great idea then and I still do. I have some minor concerns as follows:

1. Description of the integration of bar-codes into the different strains is not precise enough and seems to be contradictory in the main text and the supplementary methods. According to the main text and Data S1, diploid strains were tagged by bar-codes (Page 7, lines 145-148). However, according to the supplementary methods, haploid strains (MaTa) were tagged by bar-codes, and then mated with MaTalpha haploid strains (Page 26, lines 582-587). Please, describe explicitly how this experiment was performed and elaborate this part in the main text.
2. The authors designed the experiment to measure each of the 14 conditions in duplicates. They used the LIMMA R package that implements Bayesian t-test to quantify the significance growth of a strain in a given condition compared to the control. Although, the used statistical test was carefully chosen to analyze a small number of replicates within a group, I doubt that two replicates are sufficient to generate reliable results from high-throughput screens. Please, justify the validity of testing conditions in duplicates and not at least triplicates, backed-up by statistical analysis.
3. Please explain why up-tags and down-tags were normalized separately. Probably because there are two types of microarrays: one that measure up-tags and another that measure down-tags. If this is the case, please state it explicitly.
4. Calculation of the R (ratio) between the control and treatment is not described in precision. Authors should state explicitly that after each replicate in the control and in each of the 14 condition were normalized and transformed to log₂ values, the six control replicates are averaged and then subtracted from each treatment (lines 607-608). This will produce two ratios that are averaged to generate the ratio (Data S2). Most of this information is not included in the methods.
5. Concerning Fig. 3B, it is not clear to us the relation between the sentence "For each UP- and DOWN-tag pair the value that was closer to zero was chosen" with the one following it (lines 727-728).

Results

1. Page 9, Line 183, please mention explicitly your significance and effect size thresholds in addition to referring the author to the methods.
2. I think the authors should be more specific in their wording. For instance the term "protein

- complex" should be only used for complexes containing more than two proteins and complexes containing two proteins should be explicitly called "binary protein complex". For instance, this imprecision could cause some confusion for figure 2. Supposedly the authors are talking about binary complexes or a pair of interacting proteins in Figure 2, while in Figure 1, the authors were talking about complexes containing more than one protein (with one protein in common).
3. Briefly explain in the main text the GO enrichment analysis that you performed and cite FuncAssociate.
 4. Fig. 3B is not mentioned in the main text.
 5. Fig. 3C, I am wondering whether the concept of protein-centric and interaction-specific complexes is not related to the concept of party and date hubs, respectively. Please, elaborate on this relationship.
 6. Line 303, I do not think that the authors should describe subnetworks resulting from protein-centric changes as highly clustered subnetworks. They could describe them as large subnetworks otherwise they should investigate the clustering coefficient of the analyzed subnetworks.
 7. The authors referred to posttranslational events and change in localization affecting dynamic interactions, but no analyses related to these events were performed in this study.

Reviewer #3:

Report on Schlecht et al. „Quantitative analysis of protein interaction network dynamics in yeast". The understanding of condition dependent interactions will be key to a better understanding of cellular functioning and malfunctioning. Though condition can refer to many things such as genomic background, cell type, PTM status, etc. the authors literary assaying yeast growth for 1377 interactions between 624 yeast proteins under 14 different the condition. The interactions are recapitulated in a DHF complementation system form the screen of Tarrasov et al.. Similar approaches have been done by the group previously and be the Landry group, maybe not at that broad scale.

They find that changes of interaction are preferentially found in subnetworks around hubs (i.e. star like structures in the network). They also find that hub protein expression is a very good predictor of interaction dynamics. Notably, a major motivation (page 2 top paragraph) of their study is referring to limitations of (seminal) papers that integrate expression profiles to learn about network dynamics in a binary fashion.

Some specific points for consideration.

*) The analysis of dynamic components as 'protein-centric' vs as 'interaction-specific' is the most interesting part, in particular if the authors would characterize the latter cases. They analyze these dynamic components through modelled networks: generated random dynamic networks with different proportions of protein-centric and interaction-specific effects, and examined which were most consistent with our observed data. They judged the size of the largest component. This is very unclear to me (to say the least). How to measure interaction-specific effects by the size of the largest component? This is not likely to work from the design point of view as interaction-specific effects will be more disconnected than hubs. Why don't they use degree controlled randomization of their data to assess what are expected changes and what are more specific ones.

*) In general I do not see how their analyses are controlled for degree! If they are not properly control the results will always be a function of the number of interactions.

*) They do not speak about affinity that cannot be uncoupled from concentration.

*) High-resolution mapping of protein concentration reveals principles of proteome architecture and adaptation. Levy ED, Kowarzyk J, Michnick SW. Cell Rep. 2014. This paper is highly related to the work and ignored. A through comparison is warranted. Does expression correlate or maybe match the data of Levy et al.?

*) Not a single yeast spot is shown. To get an idea for how a conditional result can look like representative plates are required. I understand that the screen is carried out in a pooled barcoded way, but individual test need to be carried out to confirm the approach. Figure SIC is not very convincing in this respect.

*) pairwise protein abundance', is actually not a pairwise protein abundance but the minimal one. This is strange and may influence the hub vs specific interaction. A concerted change of a protein with all its interaction partner should then be seen if it is the minimum expression of a single protein? What about involvement in different function? What about different KDs.

*) They test for GO term enrichment of accumulated and depleted interactions. They "harbored" a higher-than-expected density. Well what is expected? Does this involve degree preserved network rewiring as interacting proteins will share GO term. The example suggests it is node enrichment: 'alpha-amino acid biosynthetic process' term under doxorubicin: three of the four complexes are homodimers!

*) the doxorubicin example used several times is bad as it is essentially homo dimers.

*) How does tagging effect protein half-life. Does the relative abundance of their proteins correlate with the mass spec abundance of yeast data?

*) "Such approaches can directly measure not only the presence or absence of a biophysical PPI, but also its dynamic dependence on growth environment." I do not think they measure the absence of an interaction!

1st Revision - authors' response

10 May 2017

Response to Referees' comments

For MSB-17-7532 "Quantitative analysis of protein interaction network dynamics in yeast"

Editorial comments:

Thank you again for submitting your work to Molecular Systems Biology. We have now heard back from the three referees who agreed to evaluate your study. As you will see below, the reviewers appreciate the broad scale at which condition-dependent protein-protein interactions are analyzed. However, they raise a number of concerns, which should be carefully addressed in a revision of the manuscript. The reviewers' recommendations are rather clear so I think that there is no need to repeat the points listed below.

On a more editorial level, we would like to ask you to address the following issues:

- Please provide individual figure files for the main figures and the EV figures.*
- We have replaced Supplementary Information by the Expanded View (EV) format. In this case, all additional Figures can be displayed as Expanded View Figures. EV Figures should be cited as 'Figure EV1', 'Figure EV2' etc. and they should be provided as separate files. Their legends should be included in the main text. For detailed instructions regarding expanded view please refer to our Author Guidelines: <<http://msb.embopress.org/authorguide#expandedview>>.*
- Datasets should be labeled and cited as Dataset EV1, Dataset EV2 etc.*
- Please include a Data Availability section at the end of the Materials and Methods, describing where newly generated data have been made available (i.e. at GEO or as EV datasets).*
- Please provide a "standfirst text" summarizing the study in one or two sentences (approximately 250 characters), three to four "bullet points" highlighting the main findings and a "synopsis image" (211x157 pixels, jpeg format) to highlight the paper on our homepage.*

- When you resubmit your manuscript, please download our CHECKLIST (<http://embopress.org/sites/default/files/Resources/EP_Author_Checklist_Master.xlsx>) and include the completed form in your submission. **Please note** that the Author Checklist will be published alongside the paper as part of the transparent process <<http://msb.embopress.org/authorguide#transparentprocess>>.

We thank the editor and reviewers for a thorough and thoughtful review process. The manuscript has been carefully revised to address the reviewer comments, as well as the editorial requirements. Below we include a point-by-point response, indicating changes made to the revised manuscript.

In addition, we have thoroughly reviewed the manuscript to ensure correctness, and have made the following additional edits not requested by reviewers:

- In Data EV1, the correct description is now given in the ReadMe for ‘PPI’ in the Regrowth sheet
- We have updated the Methods to have a more complete description of how strains were selected for barcoding
- We have updated numbers in the text to reflect the fact that a small number of PPIs were barcoded twice, e.g., 1379 interactions were measured amongst 1383 strains rather than 1383 interactions, 757 dynamic complexes measured in 758 strains rather than 758 dynamic complexes.
- In Data EV2 Sheet 5, (“Excluded –MTX”), we have altered the descriptions of complex change to match the manuscript (‘enhanced’ → ‘accumulated’, ‘diminished’ → ‘depleted’), and added ‘by association’ which identifies strains excluded because they carry PCA fragments detected in multiple other excluded strains. These are also now described in the Methods
- We realized that some strains in Data EV2 Sheet 5 that were marked as excluded, were inadvertently included in previous analyses. The revised manuscript removed these strains, leading to minor changes in results:
 - 1 binary complex previously found altered in the AA-mixture condition has been removed
 - minor changes in the list of enriched/depleted GO terms, none affecting a GO term discussed in the main text
 - minor differences in simulation results (40-100% protein-centric changes most consistent with component sizes instead of 50-100%; 10% interaction-specific changes required for subgraph density consistency instead of 10-20%), which did not affect conclusions
 - minor differences in the expression-PCA results, not affecting conclusions (88% instead of 87% accurate when predicting direction of concerted hub changes; $p=4.8e-05$ instead of $7e-05$ in prediction accuracy for concerted vs non-concerted hub changes, 34% estimated variance explained in all predictions instead of 33%; 75% and 86% precision for predicting direction of accumulated and depleted complexes at ($|\log_2(R)| > 2$) instead of 74% and 83%).
- Strains with a lack of barcode correspondence ($|\log_2(R_{UPtag}) - \log_2(R_{DOWNtag})| > 1$) had been excluded in Fig. 4 and EV4. Furthermore, genes with no mRNA expression measurements in either

condition were treated as unchanged in the mRNA-based modeling. This is now stated in the Methods

- The description for Figure EV1A has been updated for clarity
- Figure EV1F and its caption has been updated to remove ambiguity in the x-axis description
- Dataset EV5 now contains the mass-action-based predictions of complex abundance changes in a diauxic shift
- We have added a citation to Mailand *et al.* 2013 (PMID 23594953) in the Introduction
- We have added a citation to Smith *et al.* 2017 (PMID 28193641) in the Results
- We have made a number of stylistic changes to the text that improve clarity

Reviewer #1:

The manuscript, titled "Quantitative analysis of protein interaction network dynamics in yeast" by Schlecht et al., describes the establishment and measurement of changes in protein-protein interactions under a series of perturbations in the budding yeast. This is an expansion built upon several previously published studies. The authors implemented a highly multiplexed murine dihydrofolate reductase (mDHFR)-based protein complementation assay to quantitatively measure formation of binary protein-protein interactions in yeast. After introducing DNA barcodes to 1383 yeast strains that each harbored a binary PPI complex, which was fused to fragment of mDHFR, the authors pooled these strains and grew them together in the presence of methotrexate (MTX) under 14 different conditions. Because formation of a binary protein complex would reconstitute mDHFR, which renders the cells resistance to MTX in a quantity-dependent manner, they measured abundance of the DNA barcodes as a surrogate of quantitative measurement of binary protein complex formation. They found that many protein complexes were environment dependent. Interestingly, hub proteins (those with many known PPIs) showed concerted changes. To better understand the biology behind these observed changes in protein complexes, they integrated mRNA expression data into a diauxic shift from glucose to ethanol and observed that mRNA levels could predict concerted changes with 87% accuracy.

This is a very well designed large-scale study of PPI changes under 14 conditions built upon the success of several previous studies on a smaller scale. Compared with Y2H approach or mapping of protein complexes using IP-MS/MS, the presented approach enables examination of changes in binary PPI complexes under physiological conditions. I do not have any major concerns but only minor ones.

We appreciate the positive feedback.

Specific comments:

Reviewer Comment 1.1

1. Can the authors elaborate on their ability to detect relatively weaker PPIs?

Response 1.1

Unfortunately, affinity data for PPIs are scarce and unstandardized with respect to critically important conditions (salt, pH etc) so it is very difficult to determine whether PCA or BC-PCA captures weaker or stronger interactions in general. An existing database (<http://www.pdbbind.org.cn/>) contains Kd information for only ~1700 protein-protein interactions across all species, with 56 PPIs between yeast proteins. Not surprisingly given the scarcity of this data, overlap was small with only 2 Kds corresponding to a single interaction in our study:

PDB ID	Gene1	Gene2	Kd
3sja	GET3	GET1	17nM
3zs8	GET3	GET1	51nM

The substantial difference between these estimates highlights that, even if Kd data were available for more pairs, we might still be limited by standardization or reliability issues with Kd evidence.

For what it is worth, we show below how the average of these two values (red line) compares with the distribution of all reported yeast PPIs in this database, but we cannot conclude anything with one example:

If, instead of considering affinity by itself, we consider complex abundance (which is a function of both affinity and concentration of the two proteins), then the mDHFR-based PCA method has been estimated to capture interactions with as few as 25 complexes per cell (PMID: 18467557). However, we identify interactions involving proteins that are generally more abundant (Fig EV1D). The manuscript has been modified to capture the above two points (lines 66-68, 145-147).

Reviewer Comment 1.2

2. Can different affinity of PPIs affect the abundance of the DNA barcodes?

Response 1.2

It has been shown in a previous study (PMID: 23099892) that PCA can indeed distinguish between Ras mutants which are known to affect the affinity of the

Ras–RBD complex, so it is reasonable to assume that affinity is a factor which would affect DNA barcode abundance in this assay as well.

Reviewer Comment 1.3

3. Have the authors detected any kinase-substrate interactions in their assays?

Response 1.3

From to the Yeast Kinase Interaction Database (PMID: 21492431), we have detected the following 19 pairs described by the database as ‘relevant to phosphorylation events’.

Gene1	Gene2	Database Evidence of Kinase-Substrate Phosphorylation
HOG1	RCK2	✓
CKB1	CKA1	X
GIN4	NAP1	X
TPK1	TPK1	✓ (autophosphorylation)
BCY1	TPK1	✓
AKR1	YCK1	X
YPD1	SSK1	✓
BEM1	STE20	X
FMP45	PKH2	X
SSK1	SSK2	X
GIN4	MID2	X
NNK1	URE2	X
HSP30	STE20	X
NAP1	RIM11	X
CKA1	CKB2	X
CKB1	CKA2	X
CKB2	CKA2	X
GIN4	PDR12	X
MCK1	PEX11	X

Reviewer Comment 1.4

4. On page 7 the authors stated that a significant correlation between growth rate and "pairwise protein abundance" was observed. Is this also true for those hub proteins?

Response 1.4

Considering only hub proteins, we found a similar correlation between growth rate and pairwise protein abundance ($r=0.37$ vs $r=0.30$; difference between correlation was insignificant at $p = 0.18$).

Reviewer #2:*Reviewer Comment 2.1*

The manuscript describes an expansion of an existing methodology that they reported fairly recently in PNAS to monitor changes in a large number of protein-protein interactions following perturbations of cells. I thought that it was a great idea then and I still do. I have some minor concerns as follows:

We appreciate the positive feedback.

1. Description of the integration of bar-codes into the different strains is not precise enough and seems to be contradictory in the main text and the supplementary methods. According to the main text and Data S1, diploid strains were tagged by bar-codes (Page 7, lines 145-148). However, according to the supplementary methods, haploid strains (MaTa) were tagged by bar-codes, and then mated with MaTalpha haploid strains (Page 26, lines 582-587). Please, describe explicitly how this experiment was performed and elaborate this part in the main text.

Response 2.1

Thank you for bringing this to our attention. The latter description is the more complete one – only MATa strains had barcodes initially, and they were mated with MAT α strains to create barcoded diploid strains. The relevant text has been clarified (lines 150-154).

Reviewer Comment 2.2

2. The authors designed the experiment to measure each of the 14 conditions in duplicates. They used the LIMMA R package that implements Bayesian t-test to quantify the significance growth of a strain in a given condition compared to the control. Although, the used statistical test was carefully chosen to analyze a small number of replicates within a group, I doubt that two replicates are sufficient to generate reliable results from high-throughput screens. Please, justify the validity of testing conditions in duplicates and not at least triplicates, backed-up by statistical analysis.

Response 2.2

Below we summarize our statistical methods and further justify the use of duplicate testing conditions.

In a standard t-test, estimating variance based on two samples in each of the two groups being compared would indeed be problematic. In this experimental setting, however, measurements are available for many tags per sample, and techniques originally developed in the context of analysis of microarray-based mRNA abundance allow one to use this as additional information to learn a general error model that informs measurements of variance for abundance in each tag.

More specifically, in a standard t-test used in the R programming language with unequal sample sizes and variances, the variance of both samples is used in both calculating the t-statistic (t) and an approximation for the degrees of freedom (v), which is used to obtain a p-value for a given t :

$$t = \frac{\bar{X}_1 - \bar{X}_2}{\sqrt{\frac{s_1^2}{N_1} + \frac{s_2^2}{N_2}}}$$

$$v \cong \frac{\left(\frac{s_1^2}{N_1} + \frac{s_2^2}{N_2}\right)^2}{\frac{s_1^4}{N_1^2(N_1 - 1)} + \frac{s_2^4}{N_2^2(N_2 - 1)}}$$

Where \bar{X}_1 and \bar{X}_2 are the sample means, s_1^2 and s_2^2 are the sample variances, and N_1 and N_2 are the sample sizes. The limitation with using small sample sizes with this test is that the estimates of s_1^2 and s_2^2 may be inaccurate, potentially leading to false positives. The use of the LIMMA package is that so instead of estimating s_1^2 and s_2^2 directly from the tag fluorescence of each strain, we may use the variances of other measured strains with similar behaviour as prior information, in effect serving to ‘regularize’ or nudge the variance estimate towards the variance generally observed for observations having a similar mean. The authors of this method provide the details of how this is performed, and importantly they show that it reduces the false positive rate for large experiments with few samples compared to a conventional t-test, both mathematically and through the use simulated data (PMIDs: 16646809 , 25605792).

We do include 6 replicates for the solvent control to allow a more robust calculation of one of these variances, and Fig EV1E shows that our fluorescence signal is highly reproducible for all 6. Similarly, the log-ratio estimates ($\log_2(R)$) are also reproducible when comparing condition to control, as is evident in the heatmap in Fig 1A. This makes it possible to achieve significance even with two replicates and multiple testing correction as the sample variances are small compared to the mean.

We have also effectively treated the ‘Up’ and ‘Down’ tags separately as internal replicates, and have required that both of them show a significant difference between conditions. The requirement for tag reproducibility serves as a way to mitigate spurious outliers generated by hybridization or PCR artifacts to which duplicate measures might otherwise be susceptible.

To empirically analyze our susceptibility to spuriously significant results, we have now taken our data and shuffled the control and treatment labels for each PPI under each tag and under each condition (i.e. all entries in the data frame are shuffled independently with respect to these labels, but the shuffled order is forced to be identical for Up and Down tag pairs of the same strain). We have

put this shuffled dataset into the identical statistical pipeline as used for our real data – with the exception of omitting the effect size threshold (which would increase stringency further, but we wished to explore significance testing specifically). We have generated 100 randomly shuffled datasets to show the distribution of falsely significant results. This number is very small compared to the number of significant complex abundance changes found in our dataset (>1400 significant results above our effect size threshold when considering all conditions), and gives us an a false discovery rate estimate of ~1%. We are happy to include these results at the discretion of the editor.

Reviewer Comment 2.3

3. Please explain why up-tags and down-tags were normalized separately. Probably because there are two types of microarrays: one that measure up-tags and another that measure down-tags. If this is the case, please state it explicitly.

Response 2.3

The Up and Down tag probes are present on the same array. However, PCR was performed separately for pools of Up and Down tags, potentially leading to systematic differences in the total amount of labeled nucleic acid material between Up and Down tag pools. Therefore, they were normalized separately. This practice is also stated in our published protocol for analyzing barcoded yeast strains with this microarray (PMID: 27587778). The text now includes this explanation (lines 617-619)

Reviewer Comment 2.4

4. Calculation of the R (ratio) between the control and treatment is not described in precision. Authors should state explicitly that after each replicate in the control and in each of the 14 condition were normalized and transformed to log2 values,

the six control replicates are averaged and then subtracted from each treatment (lines 607-608). This will produce two ratios that are averaged to generate the ratio (Data S2). Most of this information is not included in the methods.

Response 2.4

The reviewer's description of the procedure is a correct description of the log₂-ratio calculation (rather than the ratio). We greatly appreciate their willingness to help us improve the manuscript at this level of detail. We have edited the column names and descriptions in Data EV2 to correct discrepancies between R and log₂(R). We have also edited lines 621-623 in order to add the above details, fixed various ambiguities between R and log₂(R) throughout the Methods, and have more precisely described how the separate log₂(R) values for the UP and DOWN tags are treated throughout the manuscript.

Reviewer Comment 2.5

5. Concerning Fig. 3B, it is not clear to us the relation between the sentence "For each UP- and DOWN-tag pair the value that was closer to zero was chosen" with the one following it (lines 727-728).

Response 2.5

We have edited lines 742-751 to clarify this. The purpose of picking the value closest to zero was simply to make the 'concerted action' test more conservative – this analysis decision had a similar motivation as our conservative choice to require a significant result for both the Up and Down tags in our initial statistical analysis.

Reviewer Comment 2.6

Results

1. Page 9, Line 183, please mention explicitly your significance and effect size thresholds in addition to referring the author to the methods.

Response 2.6

We have corrected lines 193-195 in order to state this explicitly.

Reviewer Comment 2.7

2. I think the authors should be more specific in their wording. For instance the term "protein complex" should be only used for complexes containing more than two proteins and complexes containing two proteins should be explicitly called "binary protein complex". For instance, this imprecision could cause some confusion for figure 2. Supposedly the authors are talking about binary complexes or a pair of interacting proteins in Figure 2, while in Figure 1, the authors were talking about complexes containing more than one protein (with one protein in common).

Response 2.7

Thank you, we have gone through the manuscript and changed many instances of "protein complex" to "binary protein complex". When referring to multiple

binary protein complexes which contain a single protein, we have stated this explicitly (e.g. “the addition of hydrogen peroxide led to the accumulation of binary complexes containing the Ftr1 protein”).

Reviewer Comment 2.8

3. Briefly explain in the main text the GO enrichment analysis that you performed and cite FuncAssociate.

Response 2.8

We have added some details about the GO enrichment algorithm to the main text (lines 238-241, 250-253), updated its description in the methods (lines 669-690), and taken the previously-missed opportunity to cite our own work.

Reviewer Comment 2.9

4. Fig. 3B is not mentioned in the main text.

Response 2.9

There was one reference to Figure 3B in line 285, but we have added another reference to Figure 3B on line 299 to highlight it in another relevant context.

Reviewer Comment 2.10

5. Fig. 3C, I am wondering whether the concept of protein-centric and interaction-specific complexes is not related to the concept of party and date hubs, respectively. Please, elaborate on this relationship.

Response 2.10

This is a very interesting suggestion that we had not considered previously. It would certainly make sense if ‘party hubs’ were more likely to exhibit concerted complex changes than ‘date hubs’. Unfortunately, the overlap of previously-defined party hubs and date hubs with the hubs in our data is low (5 and 1, respectively). However, we have added a mention of party and date hubs in the discussion (lines 493-498) as an interesting direction for future inquiry.

Reviewer Comment 2.11

6. Line 303, I do not think that the authors should describe subnetworks resulting from protein-centric changes as highly clustered subnetworks. They could describe them as large subnetworks otherwise they should investigate the clustering coefficient of the analyzed subnetworks.

Response 2.11

We agree, and have edited line 304 to correct our descriptions accordingly.

Reviewer Comment 2.12

7. The authors referred to posttranslational events and change in localization affecting dynamic interactions, but no analyses related to these events were performed in this study.

Response 2.12

We have removed the reference to post-translational modification and localization in the abstract (lines 42-44), and have edited another mention to make clear that our study did not explore these mechanisms of differential interaction (lines 422-423).

Reviewer #3:

Report on Schlecht et al. „Quantitative analysis of protein interaction network dynamics in yeast“.

The understanding of condition dependent interactions will be key to a better understanding of cellular functioning and malfunctioning. Though condition can refer to many things such as genomic background, cell type, PTM status, etc. the authors literary assaying yeast growth for 1377 interactions between 624 yeast proteins under 14 different the condition. The interactions are recapitulated in a DHF complementation system form the screen of Tarrasov et al.. Similar approaches have been done by the group previously and be the Landry group, maybe not at that broad scale.

They find that changes of interaction are preferentially found in subnetworks around hubs (i.e. star like structures in the network). They also find that hub protein expression is a very good predictor of interaction dynamics. Notably, a major motivation (page 2 top paragraph) of their study is referring to limitations of (seminal) papers that integrate expression profiles to learn about network dynamics in a binary fashion.

Some specific points for consideration.

Thank you for the feedback. Below, we respond to each of the reviewer's specific points, but have changed the order and grouping to improve the clarity of our response.

Reviewer Comment 3.1

**) The analysis of dynamic components as 'protein-centric' vs 'interaction-specific' is the most interesting part, in particular if the authors would characterize the latter cases. They analyze these dynamic components through modelled networks: generated random dynamic networks with different proportions of protein-centric and interaction-specific effects, and examined which were most consistent with our observed data. They judged the size of the largest component. This is very unclear to me (to say the least). How to measure interaction-specific effects by the size of the largest component? This is not likely to work from the design point of view as interaction-specific effects will be more disconnected than hubs.*

Response 3.1

We sought to choose measures of network topology that should vary depending on whether or not network dynamics change in a concerted fashion around particular proteins. We certainly agree that there are many graph properties that might change depending on the extent of concerted dynamics. We have

added further justification for use of both the size of the largest component and interaction density in the dynamic subnetwork as measurable network characteristics that should depend on the extent of concerted dynamics, both on the conceptual level and by clarifying that the simulation results bear out this choice (lines 310-320, 327-335). We have also updated the definition for the 'largest component' in the methods to improve precision and clarity (lines 720-722).

Reviewer Comment 3.2

Why don't they use degree controlled randomization of their data to assess what are expected changes and what are more specific ones.

Reviewer Comment 3.3

**) In general I do not see how their analyses are controlled for degree! If they are not properly control the results will always be a function of the number of interactions.*

Response 3.2 and 3.3

We agree that our simulation results do not tell us specifically which of the observed dynamic complexes are the result of interaction-specific compared to protein-centric effects (we do address these points however – especially in Fig. 3B and Fig. 4). We have now made this point explicit in lines 336-339. Here we just wish to estimate what proportion of protein-centric versus interaction-specific changes are consistent with the data.

One of the main predictions is that protein-centric dynamics would more frequently connect multiple interaction changes to a single protein. We note that all dynamic interactions are drawn from the set of all measured BC-PCA interactions, and are hence all drawn from a network with the same degree distribution. However, we would not want to preserve degree in the sampled dynamic network given that protein-centric and interaction-specific dynamic mechanisms would be expected to impact the degree distribution. In other words, if we were to fix degree distribution for the dynamic network, we would erase an important signature distinguishing these two mechanisms and the simulations would no longer yield an accurate reflection of what patterns of interaction changes are expected under each scenario. The randomization procedure is now more clearly described (lines 691-693).

Reviewer Comment 3.5

**) High-resolution mapping of protein concentration reveals principles of proteome architecture and adaptation. Levy ED, Kowarzyk J, Michnick SW. Cell Rep. 2014. This paper is highly related to the work and ignored. A through comparison is warranted.*

Response 3.5 (Response 3.4 is grouped with Response 3.8)

Thank you for bringing this paper to our attention. Although this paper does not measure dynamic interactions between two endogenous proteins under different conditions as we do in our study, it does use components of the PCA

method to measure protein concentration in two different cellular compartments in a way that, like our assay, depends on mass-action phenomena. It then compares the differences in concentration in these two compartments and notes that they are larger than the changes in protein expression expected under two conditions (at least in their YPD vs minimal medium example). The most relevant point of comparison with this work is that the changes they observed in single protein abundances between cellular locations are larger than the changes we observed in binary protein complex abundances between growth environments. We show this below for the glucose vs. ethanol environmental change (for both observed and mRNA-predicted binary protein complex abundance).

Although the nature of what is being measured in their study and ours is essentially different (protein concentration in different compartments vs binary protein complex abundance changes in different conditions), we have added a reference to this paper as part of our discussion of the potential effects of protein localization on complex levels (lines 505-507).

Reviewer Comment 3.6

Does expression correlate or maybe match the data of Levy et al.?

Response 3.6

Below we have taken our mRNA data in the baseline condition and correlated it to the protein concentrations in the cytoplasm and plasma membrane reported by Levy et al.

The extent of agreement seen here is consistent with previous studies measuring correlation between mRNA and protein levels in yeast (PMID: 14562106).

Reviewer Comment 3.7

*) Not a single yeast spot is shown. To get an idea for how a conditional result can look like representative plates are required. I understand that the screen is carried out in a pooled barcoded way, but individual test need to be carried out to confirm the approach. Figure S1C is not very convincing in this respect.

Response 3.7

We appreciate the concern that small scale validation and illustrative examples are key for any large scale study such as this. We agree that this is not addressed in Fig S1C/EV1C, which just compares baseline growth of individually cultured strains to protein abundance. Perhaps the reviewer meant S1F/EV1F, which partially addresses this, but still just compares baseline growth of pooled/barcoded strains to individually cultured strains, and not relative growth in different environments of pooled vs individual strains.

We have, however, measured the liquid growth of many individual strains representing accumulated, depleted, and unchanged interactions in four of the tested conditions, both with and without MTX selection. These data are presented in Figure S2A/EV2A. We carried out liquid growth assays and not spot assays on solid plates because the pooled assay was performed in liquid media, and thus liquid-growth measurements of individual strains served as a more direct comparison. For example, chemical inhibitors often need to be added at higher concentrations in agar plates to attain a comparable level of growth inhibition. Indeed, in some cases, the amount of chemical inhibitor required is prohibitive in terms of cost or availability. Our validation of individual strains in

100ul liquid cultures addresses these limitations. It is also the same method we employed in a previous study (PMID: 22615397).

Reviewer Comment 3.8

**) pairwise protein abundance', is actually not a pairwise protein abundance but the minimal one. This is strange and may influence the hub vs specific interaction. A concerted change of a protein with all its interaction partner should then be seen if it is the minimum expression of a single protein? What about involvement in different function? What about different KDs.*

Reviewer Comment 3.4

**) They do not speak about affinity that cannot be uncoupled from concentration.*

Response 3.8 and 3.4

We defined “pairwise protein abundance” to be the lowest protein concentration for each given protein pair. We agree that the rationale for this will not be immediately obvious to the reader, so that we have added further explanation in the revised manuscript (Lines 143-145).

The rationale for using the minimum abundance of the two proteins as a first order approximation of their binary complex abundance is that, in the case where affinity is high and each pair of interacting proteins is independent of others, the abundance of the protein complex corresponds to that of the least abundant participant. The mathematical derivation of this result is now included in the methods (lines 800-816), and we thank the reviewer for pointing out that it was unclear previously. We note that the original manuscript had a typographical error in an equation used in these lines, which has been corrected in the revised manuscript. We also confirmed that the corresponding code had used the correct equation all along.

To address point 3.4, we agree that in general we cannot uncouple the effects of affinity from those of concentration (see response 1.2), and that affinity is likely a factor that diminishes the correlation of our predicted abundances with growth rate. That said, K_d values are largely unavailable for PPI data (see response to reviewer comment 1.1). We have instead employed an approach from a previous study exploring the effects of concentration changes in PPI networks (PMID: 17699619). Specifically, we assigned a K_d to each interaction which is low compared to the concentration of the two interacting proteins ($\max(C_{P1}, C_{P2}) / 20$). This value was stated incorrectly in the text as $\min(C_{P1}, C_{P2}) / 20$ and we have corrected it in line 808. We also confirmed that the original implementation used the correct equation. Thus, we effectively modeled each interaction to be relatively strong. We have modified the text to make this ‘strong interaction’ modeling decision more explicit (lines 374-375).

Below we compare correlations of growth rate using binary protein complex abundance predictions under the “minimum protein abundance” model (left plot) and under the model that explicitly considers modeled K_d values (right plot). The results are nearly indistinguishable.

We completely agree that, assuming strong pairwise interactions, changing the expression of a hub will have a concerted effect on the interactions of which it is the less abundant partner. Indeed, this fits with the fact that we found many dynamic interactions to be concerted around ‘hub proteins’ that are changing in expression level in a manner consistent with the interactions in which they participate.

Reviewer Comment 3.9

**) They test for GO term enrichment of accumulated and depleted interactions. They "harbored" a higher-than-expected density. Well what is expected? Does this involve degree preserved network rewiring as interacting proteins will share GO term. The example suggests it is node enrichment: 'alpha-amino acid biosynthetic process' term under doxorubicin: three of the four complexes are homodimers!*

Reviewer Comment 3.10

**) the doxorubicin example used several times is bad as it is essentially homo dimers.*

Response 3.9 and 3.10

With hindsight, we agree with the reviewer that not specially treating homodimers in this analysis made the interpretation of ‘within-GO-term interaction density enrichment’ unclear and prone to the observed homodimer enrichment (although still statistically valid). We have opted to remove this analysis from the paper, as we found that simply excluding homodimers from the analysis results in terms largely overlapping with those in Figure 2C and S3A/EV3A. Figure S3B/EV3B have now been removed along with all their mentions in the text.

Reviewer Comment 3.11

**) How does tagging effect protein half-life. Does the relative abundance of their proteins correlate with the mass spec abundance of yeast data?*

Response 3.11

In the previously mentioned Levy et al. paper (Cell Reports 2014), protein abundance was measured for proteins with the same PCA F[3] tag that we used in our study, and their results correlated well with endogenous untagged absolute protein abundance measured by mass spectrometry (deGodoy et al. Nature 2008), suggesting that the tag does not generally have a major impact on protein stability.

Reviewer Comment 3.12

**) "Such approaches can directly measure not only the presence or absence of a biophysical PPI, but also its dynamic dependence on growth environment." I do not think they measure the absence of an interaction!*

Response 3.12

We agree and have revised this sentence (lines 66-68).

2nd Editorial Decision

12 June 2017

Thank you again for submitting your work to Molecular Systems Biology. We have now heard back from the two referees who were asked to evaluate your study. As you will see below, both reviewers are satisfied with the modifications made and they think that the study is now suitable for publication.

Before we formally accept the manuscript for publication we would like to ask you to fix a couple of remaining editorial issues listed below.

REFeree REPORTS

Reviewer #2:

Thank you to the authors for their efforts to provide detailed and informative responses to my comments.

I have read all responses to the three reviewers comments and the revised manuscript.

I am satisfied with the authors' responses and I recommend publication.

Reviewer #3:

Revised version „Quantitative analysis of protein interaction network dynamics in yeast".
The authors discussed the points raised, though most of it remained in the rebuttal letter. I still do not get the point why degree preserved randomization should kill the dynamic signature. There are dynamic links and less dynamic links which will be newly distributed in the random model to proteins which keep the number of links. Why should the degree of the protein change in the null model, as the dynamic signal is measured as function of a link? The danger of this analysis is that the dynamics observed may be a trivial function of the degree as the null model will be very very different from the real data. Anyway, my task / role as a reviewer is to promote science and not to prevent science and in general the work is interesting and of high quality. I suggest to go ahead.

Corresponding Author Name: Robert P. St.Onge and Frederick P. Roth

Manuscript Number: 7532